# O-GlcNAcylation is required for B cell homeostasis and antibody responses

Jung-Lin Wu [1], Ming-Feng Chiang[1], Pan-Hung Hsu[2], Dong-Yen Tsai[1,3], Kuo-Hsuan Hung[1], Ying-Hsiu Wang[1,4], Takashi Angata[5] & Kuo-I Lin [1]

O-linked N-acetylglucosamine (O-GlcNAc) transferase (Ogt) catalyzes O-GlcNAc modification. O-GlcNAcylation is increased after cross-linking of the B-cell receptor (BCR), but the physiological function of this reaction is unknown. Here we show that lack of Ogt in B-cell development not only causes severe defects in the activation of BCR signaling, but also perturbs B-cell homeostasis by enhancing apoptosis of mature B cells, partly as a result of impaired response to B-cell activating factor. O-GlcNAcylation of Lyn at serine 19 is crucial for efficient Lyn activation and Syk interaction in BCR-mediated B-cell activation and expansion. Ogt deficiency in germinal center (GC) B cells also results in enhanced apoptosis of GC B cells and memory B cells in an immune response, consequently causing a reduction of antibody levels. Together, these results demonstrate that B cells rely on O-GlcNAcylation to maintain homeostasis, transduce BCR-mediated activation signals and activate humoral immunity.

[1] Genomics Research Center, Academia Sinica, No. 128, Section 2, Academia Road, Nankang District, Taipei 115, Taiwan. [2] Department of Bioscience and Biotechnology, National Taiwan Ocean University, No. 2, Beining Road, Jhongjheng District, Keelung 202, Taiwan. [3] Institute of Biochemistry and Molecular Biology, National Yang-Ming University, No. 155, Section 2, Linong Street, Beitou District, Taipei 112, Taiwan. [4] Graduate Institute of Life Sciences, National Defense Medical Center, No. 161, Section 6, Minquan East Road, Neihu District, Taipei 114, Taiwan. [5] Institute of Biological Chemistry, Academia Sinica, No. 128, Section 2, Academia Road, Nankang District, Taipei 115, Taiwan. Jung-Lin Wu and Ming-Feng Chiang contributed equally to this work. Correspondence and requests for materials should be addressed to T.A. (email: angata@gate.sinica.edu.tw) or to K.-I.L. (email: kuoilin@gate.sinica.edu.tw)

Post-translational modifications (PTM) are important for modulating the function of proteins. In addition to phosphorylation, acetylation and methylation, O-linked N-acetylglucosamine (O-GlcNAc) modification is an important PTM of intracellular proteins[1] and is essential for mouse embryonic development[2]. O-GlcNAcylation is catalyzed by O-GlcNAc transferase (Ogt) and is the addition of a GlcNAc unit to serine or threonine residues of nuclear and cytosolic proteins, and can be reversibly removed by O-linked N-acetylglucosaminidase (Oga)[3]. O-GlcNAcylation regulates many protein functions, including protein kinase activity, DNA-binding activity, protein–protein interaction, localization and stability, and has been implicated in diseases, including neurodegenerative diseases, diabetes, and cancers[4, 5]. Many proteins involved in neurodegenerative diseases, such as Alzheimer's disease-related tau protein and amyloid precursor protein, are modified by O-GlcNAcylation, the reduction of which is associated with disease progression[6, 7]. Protein O-GlcNAcylation has also been shown to control T-cell progenitor renewal and malignancy[8].

The maintenance of B-cell homeostasis depends on two central signals from the B-cell activating factor (BAFF) receptor (BAFFR) and B-cell receptor (BCR)[9, 10]. BAFF belongs to the TNF family, and is also known as TNF superfamily member 13B (TNFSF13B)[9]. Deletion of the gene encoding BAFF or BAFFR in mice results in the loss of the majority of peripheral mature B cells[11, 12]. BCR signaling is not only crucial for antigen-dependent activation of B cells, but also important for promoting survival mediated by ligand-independent tonic signals[13]. The PI3K/AKT signaling cascade has a critical function in the control of BCR-dependent mature B-cell survival[10]. After BCR ligation with antigen, phosphorylation of CD79a and CD79b, and the three main kinases downstream of BCR signaling (Syk, the Src family kinase Lyn, and the Tec-family kinase Btk) transduce crucial signaling events for the activation of B cells[14].

Given that BCR-antigen engagement initiates a B-cell proliferation accompanied by rapid increases in glucose uptake and glycolysis[15], which results in the generation of UDP-N-acetylglucosamine (UDP-GlcNAc) via the hexosamine biosynthetic pathway (HBP), and that UDP-GlcNAc is the donor substrate of Ogt[16], we propose that O-GlcNAcylation may contribute to the function of B cells. Indeed, enhanced O-GlcNAcylation of transcription factors NF-κB p65 and NFATc1 by overexpression of Ogt in B or T cells results in increased cell activation[17]. Furthermore, our previous phosphoproteome-based analysis showed that phosphorylation of several proteins involved in BCR signaling was altered by the changes in protein O-GlcNAcylation[18].

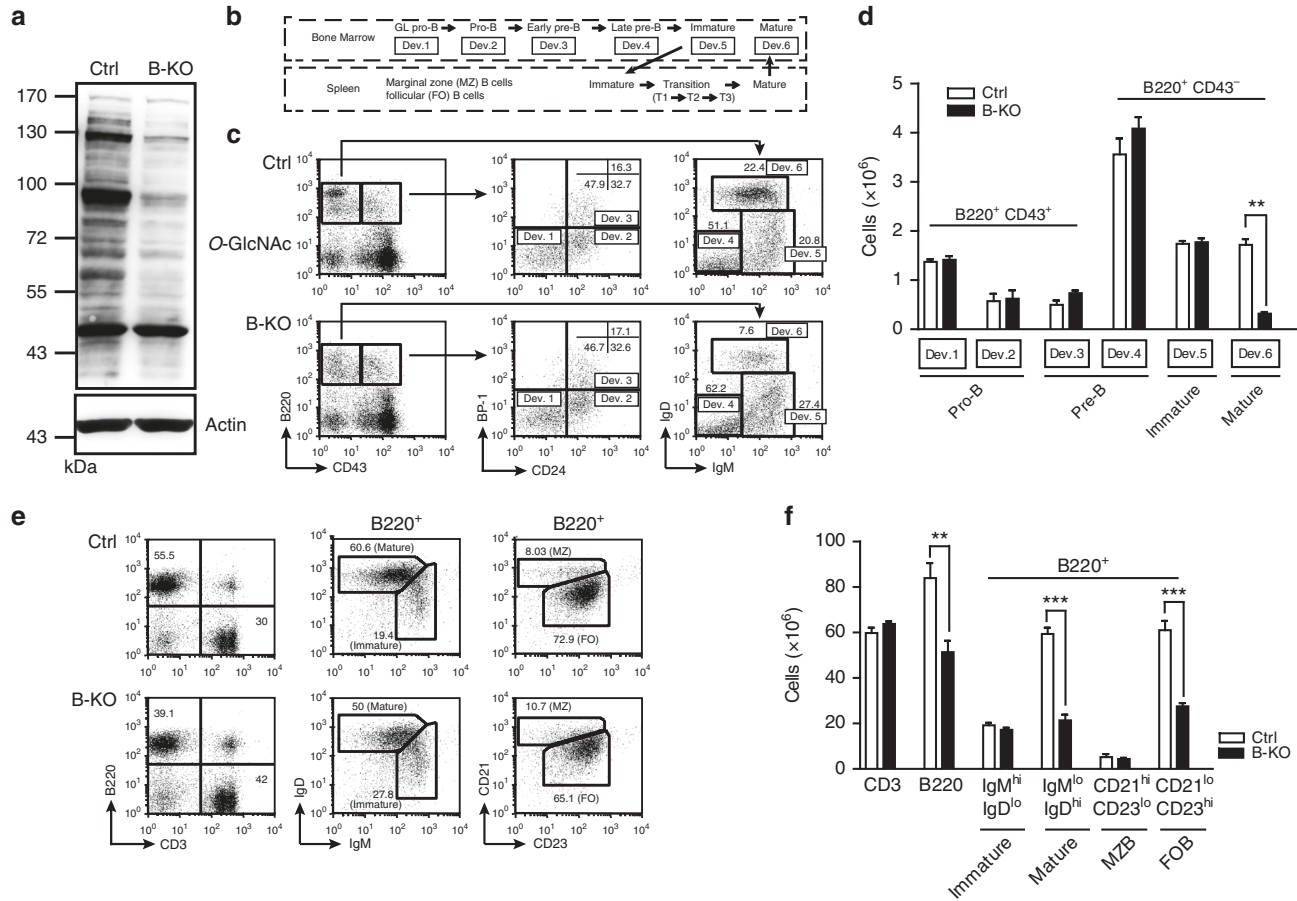

**Fig. 1** Deletion of *Ogt* during B-cell development results in reduced mature B cells. **a** The levels of *O*-GlcNAcylation in Ctrl and B-KO splenic B cells were verified by immunoblotting using anti-*O*-GlcNAc antibody. Actin was used as the protein loading control. **b** Flow chart showing the developmental stages of B cells in bone marrow and spleen as used to examine the frequency of B cells in Ctrl and B-KO mice. GL: germline. **c** Bone marrow single-cell suspensions from Ctrl and B-KO mice were subjected to flow cytometric analysis to quantify the frequency of the cells in various B-cell developmental stages. **d** Bar graphs showing the number of various B-cell subsets in Ctrl and B-KO mice during development in bone marrow. **e** Flow cytometric analysis showing the frequency of B-cell subsets in the spleen of Ctrl and B-KO mice. **f** Bar graphs showing the number of B-cell subsets in Ctrl and B-KO spleens. The numbers shown in dot plots in **c** and **e** are the percentages in quadrants or gates. The results in **d** and **f** are the mean ± s.e.m. ($n = 3$). **$P < 0.01$, ***$P < 0.001$ (two-tailed unpaired $t$ test)

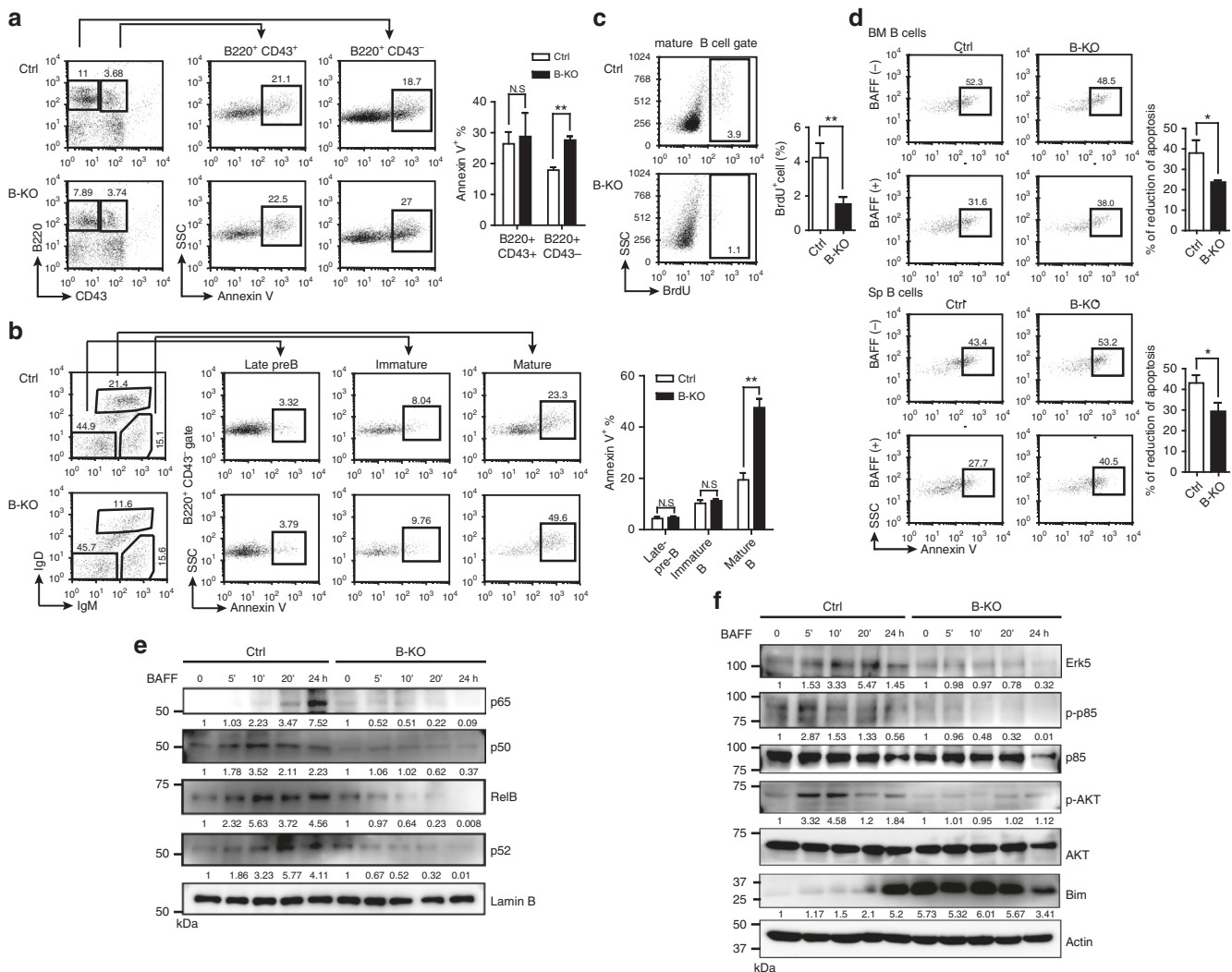

**Fig. 2** O-GlcNAcylation in BAFFR signaling regulates the survival of mature B cells. **a**, **b** Frequency of Annexin V[+] cells at various B-cell developmental stages, including pro-B (B220[+] CD43[+]) and non-pro-B (B220[+] CD43[−]) B cells (**a**), as well as late pre-B, immature and mature B cells within non-pro-B (B220[+] CD43[−]) gate (**b**), in bone marrow of Ctrl and B-KO mice. **c** Frequency of BrdU[+] mature B cells in Ctrl and B-KO bone marrow. **d** Frequency of Annexin V[+] B220[+] bone marrow and splenic B cells cultured for 72 h with (+) or without (−) recombinant BAFF (rBAFF, 200 ng ml[−1]). Percentage of reduction of apoptosis in B220[+] bone marrow and splenic B cells after BAFF treatment was calculated by (% of apoptosis in untreated group − % of apoptosis in rBAFF-treated group) / (% of apoptosis in untreated group). **e** Immunoblot of nuclear extracts isolated from rBAFF-treated Ctrl and B-KO splenic B cells at various time points with indicated antibodies against NF-κB subunits. Lamin B: nuclear protein loading control. **f** Immunoblot showing the activation of Erk5, p85/PI3K and Akt as well as Bim expression after rBAFF treatment in Ctrl and B-KO splenic B cells at various time points. Actin: protein loading control. The representative data from one of at least three independent experiments are shown. Quantification of results is shown in the right panel in **a**–**d**. The data are the mean ± s.e.m. ($n = 3$, **a**–**d**). *$P < 0.05$, **$P < 0.01$ (two-tailed unpaired $t$ test). In **e** and **f**, the quantification of band intensity is indicated. N.S., not significant

However, the function of protein O-GlcNAcylation and Ogt in the B-cell lineage in vivo is unclear. Herein, we use genetically modified mouse models in which Ogt is deleted at different stages of B-cell development to study function in the B-cell lineage. We show that Ogt is not only important for triggering BCR-mediated signaling pathways, but also for the survival of mature, germinal center (GC) and post-GC B cells in vivo. A crucial role for O-GlcNAcylation in IgG responses is also identified.

## Results

**Ogt is required for the homeostasis of mature B cells**. To study whether Ogt has a function in B cells in vivo, we generated a mouse line in which Ogt can be deleted from pre-B cells, hereafter called B-KO, by crossing Ogt-floxed mice (Ogt[f/f] or Ogt[f/Y] as Ogt

is a X-linked gene[19]) with mice expressing Cre recombinase controlled by the CD19 promoter (CD19-Cre)[20]. Deletion of Ogt is B-cell-specific as demonstrated by the intact Ogt allele in splenic CD3[+] T cells (Supplementary Fig. 1a). In splenic B cells of B-KO mice, mRNA (Supplementary Fig. 1b) and protein (Supplementary Fig. 1c) levels of Ogt were effectively decreased in comparison to littermate control (Ctrl) mice. Accordingly, overall protein O-GlcNAcylation was decreased in B-KO splenic B cells as compared with Ctrl splenic B cells (Fig. 1a). Bone marrow B220[+] B cells in B-KO mice also revealed efficient Ogt allele deletion (Supplementary Fig. 1d) and reduced mRNA (Supplementary Fig. 1e). As Ogt is not completely deleted, we only used male mice in experiments to minimize the issue of mosaicism.

To examine whether absence of Ogt in B cells affects B-cell development, we analyzed the distribution and cellularity of B-

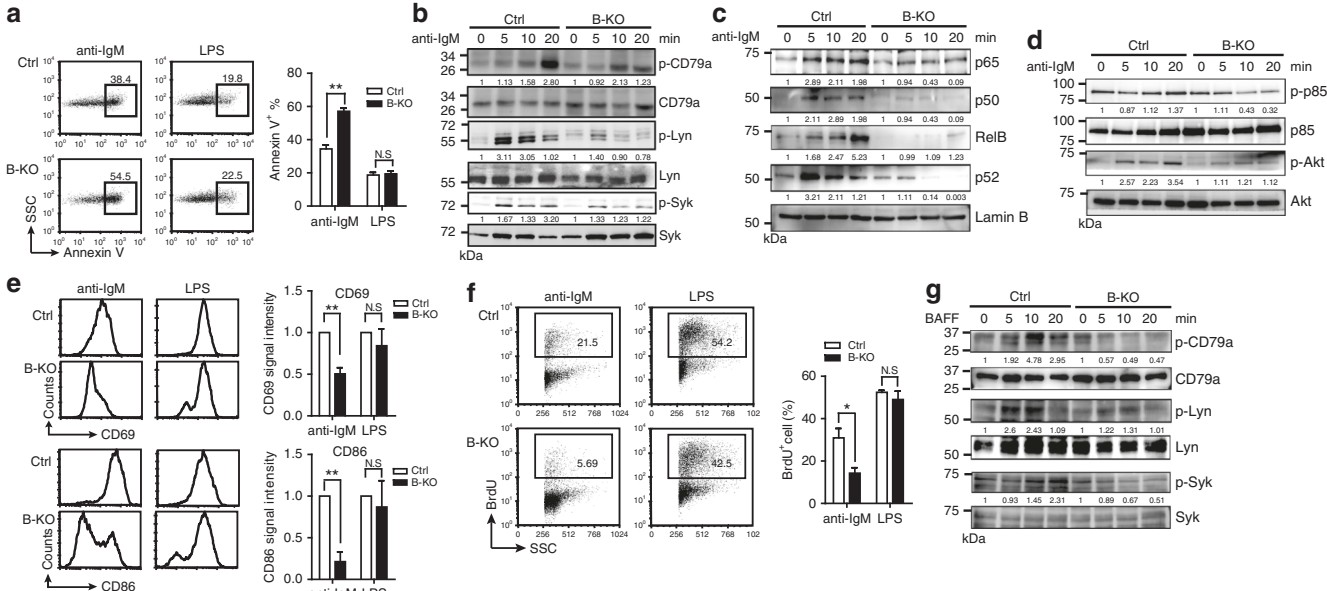

**Fig. 3** Mature B cells lacking *Ogt* do not activate after BCR cross-linking. **a** Flow cytometric analysis showing the frequency of Annexin V[+] cells in Ctrl and B-KO splenic B cells stimulated with anti-IgM (10 μg ml[−1]) or LPS (2 μg ml[−1]) for 24 h. **b** Immunoblot showing the levels of CD79a, Lyn and Syk activation in Ctrl and B-KO splenic B cells after anti-IgM stimulation. **c** Immunoblot showing the nuclear translocation of NF-κB subunits in Ctrl and B-KO splenic B cells after anti-IgM stimulation. Lamin B: nuclear protein loading control. **d** Immunoblot showing the activation of PI3K and Akt in Ctrl and B-KO splenic B cells after anti-IgM stimulation. **e** Flow cytometric analysis showing the levels of CD69 and CD86 in Ctrl and B-KO splenic B cells stimulated with anti-IgM or LPS for 24 h. Bar graphs on the right show the relative mean fluorescent intensity of CD69 (top panel) or CD86 (bottom panel) normalized by that of Ctrl B cells. **f** Flow cytometric analysis showing the frequency of BrdU[+] cells in Ctrl and B-KO splenic B cells stimulated with anti-IgM or LPS for 48 h. **g** Immunoblot showing the phosphorylation and total levels of Lyn, Syk and CD79a after rBAFF (200 ng ml[−1]) treatment in Ctrl and B-KO splenic B cells. The representative data from one of at least three independent experiments (two independent experiments in **b**) are shown. Quantification results shown in the right panel in **a**, **e** and **f** are the mean ± s.e.m. (*n* = 3). \*P < 0.05, \*\*P < 0.01 (two-tailed unpaired *t* test). In **b**, **c**, **d** and **g**, quantification of band intensity is relative to Lamin B (**c**) or each corresponding total protein (**b**, **d** and **g**) as indicated. N.S., not significant

cell subsets in Ctrl and B-KO mice using the criteria described in Fig. 1b[21]. We found that both frequency and cell number of mature B cells, but not other stages of B cells, were largely decreased in the bone marrow of B-KO mice in comparison to Ctrl mice (Fig. 1c, d). Given that mature B cells in bone marrow are re-circulating B cells, we also examined the B-cell subsets in the spleen and found that the frequency and numbers of total B220[+] B cells were decreased (Fig. 1e, f). The decreased number of splenic B cells appears to be due to the decreased number of mature B cells, particularly follicular (FO) B cells (Fig. 1e, f) but not other B-cell subsets, including immature B cells, transitional B cells (T1, T2 and T3), and marginal zone (MZ) B cells (Fig. 1f and Supplementary Fig. 2a, b).

**Mature B cells in B-KO mice undergo increased apoptosis.** In theory, if the reduced number of mature B cells is due to the developmental block, the number of immature or transitional B cells should be increased. However, B-KO mice showed normal number of immature and transitional B cells. We thus speculated that the reduced number of mature B cells in B-KO mice may result from enhanced apoptosis, which could reflect the insufficient acquisition of survival signals from the microenvironment. Therefore, we analyzed the frequency of apoptotic cells by Annexin V staining. In bone marrow, Annexin V[+] cells were increased among B220[+] CD43[−] non-pro-B cells, including late pre-B, immature B and mature B cells, in B-KO mice (Fig. 2a). This increased apoptosis was prominent in re-circulating mature B cells, but not late pre-B or immature B cells, in which the *Cd19* promoter-driven cre-recombinase should be also expressed (Fig. 2b). We next examined if the turnover rate of mature B cells in B-KO bone marrow is altered by injecting Ctrl and B-KO mice

with BrdU and subsequently measuring the frequency of BrdU incorporation in bone marrow mature B cells 12 h later. The results showed that, compared with Ctrl mature B cells, B-KO mature B cells displayed a reduced frequency of BrdU[+] cells (Fig. 2c). These combined data suggest that mature B cells need *O*-GlcNAcylation to maintain homeostasis. Furthermore, we also analyzed the apoptosis of various B-cell subsets in the spleen, including immature and mature B cells, FO B cells, MZ B cells, and transitional B cells. Only mature B cells, but not the other B-cell subsets, showed slightly augmented apoptosis (Supplementary Fig. 2c–f).

Given that the survival of mature B cells depends on BAFF-mediated signalling[9], we next examined if the frequency of apoptosis differs in B cells from the bone marrow and spleen of Ctrl and B-KO mice when stimulated with recombinant BAFF (rBAFF). We found that rBAFF treatment of freshly isolated bone marrow or splenic B cells from Ctrl mice resulted in a more significant reduction in apoptosis compared with B-KO B cells (Fig. 2d). This finding suggests that the BAFF signaling pathway may be perturbed in the absence of *Ogt*. BAFFR-mediated signaling activates several survival pathways including the activation of non-canonical NF-κB p52/RelB[22], canonical NF-κB p50/RelA[23], PI3K/AKT[24, 25] and ERK5 MAPK kinase[26] pathways. We found markedly reduced nuclear translocation of p50, RelA (p65), RelB and p52 in B-KO splenic B cells treated with rBAFF (Fig. 2e, Supplementary Fig. 3a). Moreover, the activation of Erk5, p85 subunit of PI3K and Akt were also impaired in rBAFF-treated B-KO B cells (Fig. 2f, Supplementary Fig. 3b). It has been shown that BAFF regulates B-cell survival by downregulation of Bim[27]. Accordingly, we found that Bim levels were dramatically enhanced in rBAFF-treated B-KO B cells

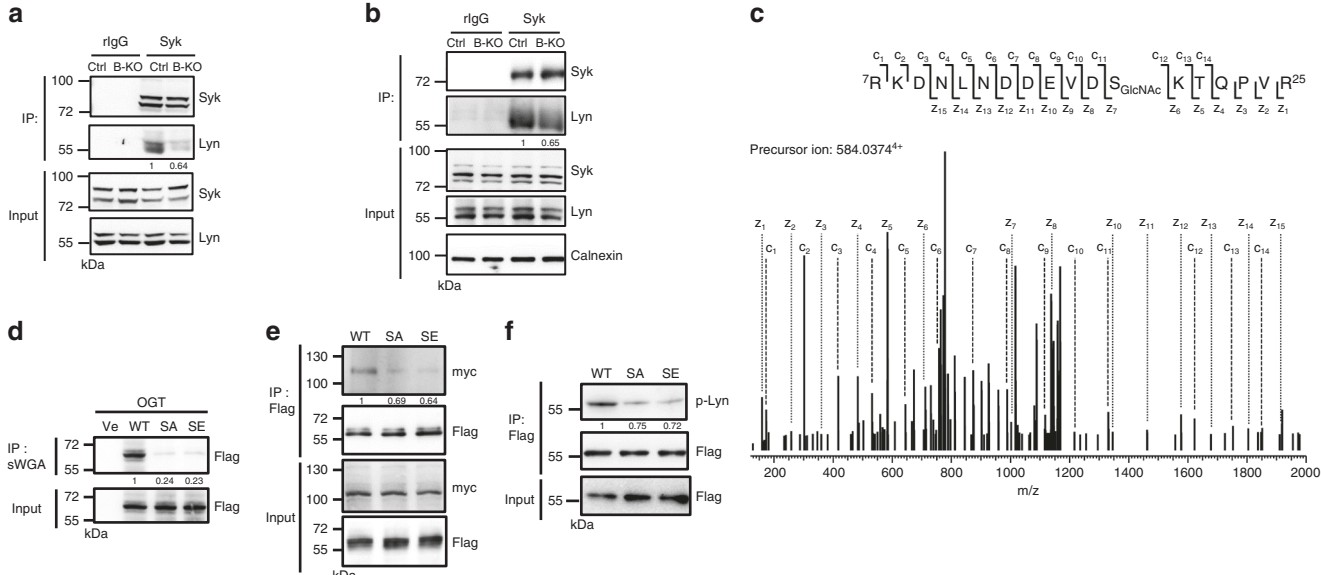

**Fig. 4** *O*-GlcNAcylation of Lyn at S19 is required for the interaction of Syk. **a** Lysates prepared from Ctrl and B-KO splenic B cells stimulated with anti-IgM for 5 min were subjected to immunoprecipitation (IP) with anti-Syk or control rabbit IgG (rIgG) antibody, followed by immunoblotting (IB) with the indicated antibodies. **b** Protein lysates from membrane fractions of anti-IgM stimulated Ctrl and B-KO splenic B cells were subjected to IP with anti-Syk or rIgG antibody, followed by IB with indicated antibodies. Calnexin served as a loading control for membrane fraction. **c** Mapping the *O*-GlcNAc site on S19 of mouse Lyn by using ETD fragmentation. The ions, $z_6^+$ (*m/z* 712.4188), $z_7^+$ (*m/z* 1002.5387), $c_{11}^+$ (*m/z* 1331.6101), and $c_{12}^+$ (*m/z* 1622.7351) indicated that S19 carried a GlcNAc moiety. **d** Lysates from 293 T cells overexpressing the vector control (Ve), Flag-EGFP-tagged WT, SA or SE Lyn were subjected to a pull-down assay using sWGA agarose beads, followed by IB with an anti-Flag antibody. **e** Lysates from 293T cells transfected with Flag-tagged WT, SA, or SE Lyn along with myc-tagged Syk were subjected to IP with anti-Flag antibody, followed by IB with the indicated antibodies. **f** Anti-Flag immunoprecipitates from Ramos B cells transfected with vector expressing Flag-EGFP-WT, -SA or -SE Lyn and stimulated with anti-IgM (25 μg ml⁻¹) for 5 min were subjected to IB with indicated antibodies. In **a**, **b**, **d**, **e** and **f**, the representative data from one of at least 2–3 experimental repeats are shown. Quantification of band intensity is indicated in **a**, **b**, **d**, **e** and **f**

(Fig. 2f). Together, this shows that BAFF-mediated mature B-cell survival is enhanced by *Ogt*.

**Ogt deficient B cells have impaired activation.** Appropriate tonic signaling via the BCR signalosome supports the survival of peripheral mature B cells[28], whereas BCR cross-linking by antigen ligation employs similar signaling pathways to activate B cells and induce apoptosis[28]. We next sought to examine if B-KO B cells are more sensitive to apoptosis after cross-linking of BCR by anti-IgM. We found that B-KO splenic B cells showed enhanced apoptosis when stimulated with anti-IgM compared with Ctrl B cells (Fig. 3a). Interestingly, the effect of protein *O*-GlcNAcylation in preventing apoptosis appears to be specific to BCR signaling because lipopolysaccharide (LPS)-induced apoptosis via Toll-like receptor 4 (TLR4) was nearly comparable between Ctrl and B-KO B cells (Fig. 3a). Thus, we wondered whether the activation of proximal BCR-mediated signaling pathways is perturbed in B-KO B cells. Indeed, we found that the phosphorylation of CD79a, Lyn, and Syk were all reduced in anti-IgM stimulated B-KO splenic B cells (Fig. 3b), which was associated with the decreased nuclear translocation of various subunits of NF-κB (Fig. 3c, Supplementary Fig. 3c). Furthermore, the activation of the PI3K/AKT pathway that transduces survival signals downstream of BCR cross-linking was also reduced in anti-IgM stimulated B-KO B cells (Fig. 3d, Supplementary Fig. 3d).

The defective activation of the BCR signaling cascades was linked with markedly reduced B-cell activation in anti-IgM stimulated B-KO splenic B cells, as evidenced by the fact that the induction of B-cell activation surface markers, CD69 and CD86, was largely abolished (Fig. 3e). Accordingly, 2 days after anti-IgM stimulation, the percentage of BrdU⁺ B-KO B cells was largely reduced as compared with stimulated Ctrl B cells (Fig. 3f). It is

noted that Ogt may selectively affect BCR pathways because upregulation of CD69 and CD86, as well as cell proliferation, appear to be not significantly altered in LPS-treated B-KO splenic B cells (Fig. 3e, f). All together, these results suggest that *O*-GlcNAcylation is required for the initiation of antigen-ligation mediated BCR signaling.

BAFFR-mediated signaling has been demonstrated to cause CD79a and Syk phosphorylation via the activation of Src kinases such as Lyn[10]. We wondered if Lyn, CD79a and Syk phosphorylation is also affected in rBAFF-treated B-KO splenic B cells. Of note, we found the phosphorylation of Lyn, which links the crosstalk between BAFFR and BCR pathways, was defective, resulting in the reduced phosphorylation of CD79a and Syk in B-KO splenic B cells stimulated with rBAFF (Fig. 3g, Supplementary Fig. 3e).

**O-GlcNAcylation of Lyn at S19 regulates Syk recruitment.** We next attempted to decipher which molecules in BCR signaling are *O*-GlcNAcylated and are important for transducing critical signals for mounting proper B-cell activation. Cell lysates prepared from splenic B cells isolated from Ctrl and B-KO mice were subjected to a pull-down assay using succinylated wheat germ agglutinin (sWGA), a lectin that has high affinity for *O*-GlcNAc[29], followed by LC-MS/MS analysis (Supplementary Fig. 4a). Comparing the proteome results from the pull-down assays with lysates from Ctrl and B-KO splenic B cells, we could identify the *O*-GlcNAcylated proteins of B cells (Supplementary Data 1). Proteins, including host cell factor 1 (HCF-1), specificity protein 1 (Sp1) and OGT, which have all previously been reported as *O*-GlcNAcylated proteins[30–32], were readily detected by sWGA pull-down. Furthermore, Ingenuity Pathway Analysis (IPA) was used to construct systematic networks of *O*-

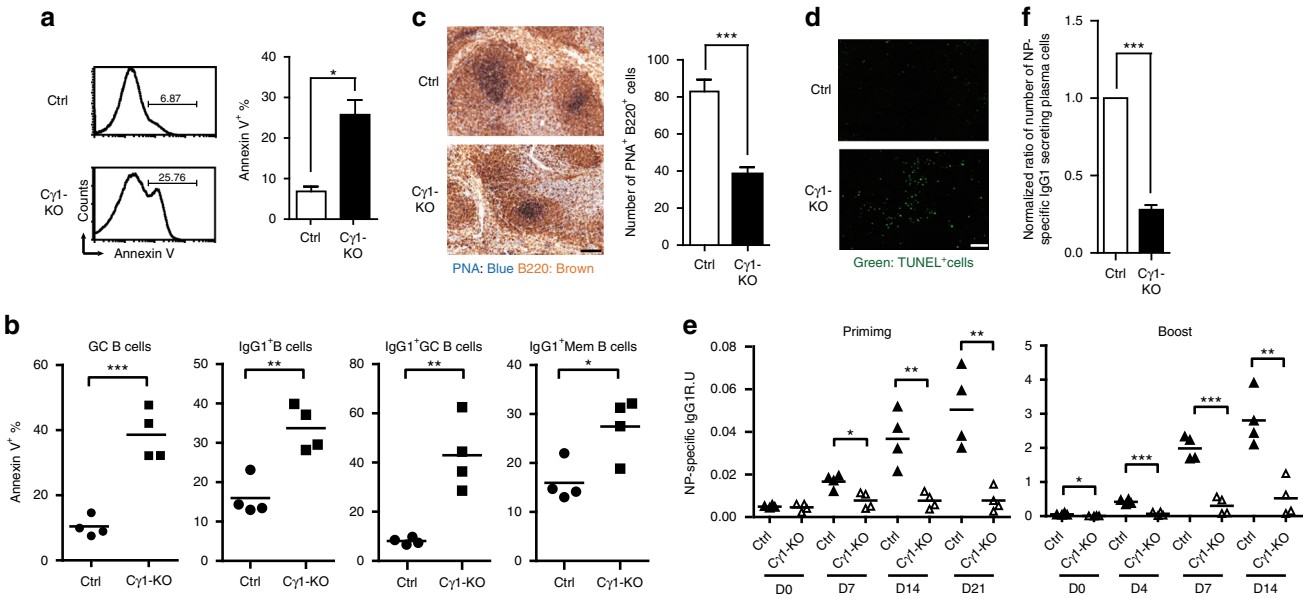

**Fig. 5** Mice lacking *Ogt* in post-GC B cells do not activate antibody responses. **a** Flow cytometric analysis of the frequency of Annexin V⁺ cells in Ctrl and Cγ1-KO splenic B cells stimulated with anti-IgM (25 µg ml⁻¹), anti-CD40 (1 µg ml⁻¹) and IL-21 (200 ng ml⁻¹) for 2 days. Quantification of results is shown in the right panel. **b** The frequency of apoptotic GC B cells (GL7⁺ CD38ˡᵒ), IgG1⁺ B cells, IgG1⁺ GC B cells (IgG1⁺ GL7⁺ CD38ˡᵒ) and IgG1⁺ memory B cells (IgG1⁺ GL7⁻ CD38ʰⁱ) were examined by Annexin V staining at 14 days after NP-KLH immunization. **c** Immunohistochemistry staining with anti-B220 and peanut agglutinin (PNA) lectin on spleen sections of Ctrl and Cγ1-KO mice 14 days after immunization. Scale bar = 100 µm. Quantification of results is shown in the right panel. **d** TUNEL assay using the embedded spleen sections from **c** showing increased apoptotic cells in immunized Cγ1-KO mice. Scale bar = 50 µm. **e** The levels of NP-specific IgG1 in Ctrl and Cγ1-KO mice after primary and secondary NP-KLH immunization at indicated days were analyzed by ELISA. **f** ELISPOT analysis of the number of plasma cells secreting anti-NP IgG1 in the bone marrow of Ctrl and Cγ1-KO mice 4 weeks after secondary NP-KLH immunization. The data in **a**, **b**, **c**, **e** and **f** are the mean ± s.e.m. (data are based on 3 pairs of mice in **a**, and 4 pairs of mice in **b**, **c**, **e** and **f**; in **c**, 5 GCs were randomly selected for counting). Representative data from one of at least two experimental repeats are shown. *P < 0.05, **P < 0.01, ***P < 0.001 (two-tailed unpaired *t* test)

GlcNAcylated proteins associated with BCR signaling. The results revealed that several proteins, including CD22, Lyn, and SHP-1 (Supplementary Fig. 4b), are relevant to BCR signaling pathways.

Given that B-KO B cells displayed severe defects in responding to anti-IgM, we suspected that the function of certain proteins proximately associated with the BCR may be affected in the absence of *O*-GlcNAcylation. Indeed, we validated that Lyn was detected in the cell lysates by sWGA pull-down at various time points before and after anti-IgM stimulation (Supplementary Fig. 4c). We also added GlcNAc together with sWGA to the lysates of unstimulated splenic B cells to demonstrate the signal specificity for *O*-GlcNAcylated proteins (Supplementary Fig. 4c). Moreover, much less Lyn was pulled-down by sWGA from B-KO B cells than from Ctrl B cells (Supplementary Fig. 4d). It is also noted that *O*-GlcNAcylation of Lyn affects its interaction with Syk because much less Lyn was co-immunoprecipitated with Syk from anti-IgM-treated splenic B-KO B cells than from Ctrl B cells (Fig. 4a). Lack of *O*-GlcNAcylation did not affect the docking of Lyn and Syk to the cell membrane of stimulated B cells because similar amounts of Lyn and Syk were detected in the membrane fraction (Fig. 4b). However, less Syk was associated with Lyn on the membrane of stimulated B-KO B cells (Fig. 4b).

We then sought to definitively map the *O*-GlcNAcylated site(s) on Lyn by mass spectrometric analysis. Vectors expressing Flag-tagged Lyn and OGT were co-transfected into human embryonic kidney (HEK) 293T cells and Lyn purified by anti-Flag was subjected to LC-MS/MS to identify *O*-GlcNAcylation sites. *O*-GlcNAcylation-modified peptides of Lyn (⁷RKDNLNDDEVDS_GlcNAc_KTQPVR²⁵) were observed at [M + 4 H]⁴⁺ m/z 584.0374, and its *O*-GlcNAcylation at S19 was

confirmed by both electron transfer dissociation (ETD) and higher collision energy dissociation fragmentation mass spectra (Fig. 4c and Supplementary Fig. 4e). A mutant Lyn, in which amino acid 19 is replaced by alanine (SA Lyn), was generated by site-directed mutagenesis. The results show that, compared with wild-type (WT) Lyn, SA Lyn expressed in HEK 293T cells failed to be pulled down by sWGA (Fig. 4d). Moreover, co-immunoprecipitation results using lysates from HEK 293T cells expressing exogenous Flag-tagged WT or SA Lyn along with myc-tagged Syk showed that, compared with WT Lyn, SA Lyn interacts poorly with Syk (Fig. 4e), supporting our notion that *O*-GlcNAcylation of Lyn is important for the interaction with Syk. Of note, in response to anti-IgM cross-linking, less phosphorylated Flag-EGFP-tagged Lyn at tyrosine 397 that represents the active form[33], was detected in transfected Ramos B cells when S19 was mutated to alanine (Fig. 4f). Since phosphorylation of Lyn at S19 was observed in phagosomes in IFN-γ-activated macrophages[34], we wondered if the less-efficient interaction between SA Lyn and Syk could result from impaired phosphorylation, rather than *O*-GlcNAcylation, at S19. To address this question, we further generated a mutant Lyn in which serine 19 was replaced with glutamic acid (SE Lyn), which acts as a phospho-mimicry mutation. Similar to SA Lyn, SE Lyn was neither pulled down by sWGA (Fig. 4d) nor co-immunoprecipitated with myc-tagged Syk (Fig. 4e), demonstrating that loss of S19 *O*-GlcNAcylation, not phosphorylation, affects Syk interaction. Moreover, SE Lyn remained less activated after anti-IgM cross-linking in transfected Ramos B cells (Fig. 4f). These combined results indicate that *O*-GlcNAcylation of Lyn at S19 promotes Lyn activation and Syk interaction.

**Ogt is required for GC B cells and plasma cells**. To study the functional role of O-GlcNAcylation in an immune response, we next crossed $Ogt^{f/f}$ or $Ogt^{f/}Y$ mice with IghCγ1-Cre mice to create a mouse line in which the Ogt allele is deleted in GC B cells[35], hereafter called Cγ1-KO mice. Splenic B cells from Ctrl and Cγ1-KO mice were stimulated with anti-IgM, anti-CD40, and IL-21, which in combination promote antibody production and class switch recombination to IgG1[36]. IgG1$^+$ cells were then sorted to demonstrate the effective ablation of Ogt in IgG1$^+$ cells of Cγ1-KO mice (Supplementary Fig. 5a). The IgG1$^+$ cells, gated from anti-IgM, anti-CD40, and IL-21 stimulated Cγ1-KO B cells, show enhanced apoptosis (Fig. 5a), suggesting that O-GlcNAcylation is also required for the survival of post-switched B cells. We then immunized Ctrl and Cγ1-KO mice with a T-cell-dependent antigen (4-hydroxy-3-nitrophenyl acetyl-keyhole limpet hemocyanin (NP-KLH)) and found that splenic B cells in Cγ1-KO mice have an increased frequency of Annexin V$^+$ apoptotic GC B cells (GL7$^{lo}$ CD38$^{lo}$) and IgG1$^+$ B cells, including IgG1$^+$ GC B cells (IgG1$^+$ GL7$^{lo}$ CD38$^{lo}$) and IgG1$^+$ memory B cells (IgG1$^+$ GL7$^-$ CD38$^{hi}$), at 2 weeks after NP-KLH immunization, as compared with Ctrl mice (Fig. 5b). This finding is linked with the reduced number of GC B cells, IgG1$^+$ GC B cells and IgG1$^+$ memory B cells in immunized Cγ1-KO mice (Supplementary Fig. 5b). Accordingly, the size and numbers of peanut agglutinin (PNA)$^+$ B220$^+$ GC B cells were reduced and the frequency of TUNEL$^+$ cells in spleens was increased in Cγ1-KO mice after NP-KLH immunization (Fig. 5c, d).

Because of decreased survival of IgG1$^+$ B cells, the levels of NP-specific IgG1 antibody were also greatly diminished in Cγ1-KO mice after primary and secondary immunization with NP-KLH (Fig. 5e). Accordingly, the numbers of long-lived plasma cells secreting NP-specific antibody in bone marrow of Cγ1-KO mice at 4 weeks after secondary NP-KLH immunization were significantly reduced as compared with Ctrl mice (Fig. 5f). Moreover, as the Cre-mediated recombination in IghCγ1-Cre mice occurs not only in IgG1 expressing B cells, but also in a fraction of B cells expressing other Ig isotypes[35], we found that the titers of NP-specific IgG2b, IgG2c, and IgG3 were all reduced after NP-KLH immunization (Supplementary Fig. 5c). These combined results suggest that O-GlcNAcylation is important for sustaining the survival of GC B cells and the production of antibodies.

## Discussion

O-GlcNAcylation modification serves as a nutrient sensor that regulates the signaling events in many physiological and diseased states[37]. In this study, we demonstrate that protein O-GlcNAcylation is not only required for mounting B-cell activation triggered by BCR ligation but also crucial for the maintenance of the survival of mature B cells, memory B cells and plasma cells.

Numerous reports demonstrate that O-GlcNAcylation plays a critical role in development and differentiation. However, reports focusing on the effects of O-GlcNAcylation on immune cell development and function are still very limited. In innate immune responses, O-GlcNAc modification of a regulatory subunit of transforming growth factor (TGF)-β-activated kinase 1 (TAK1), TAK1-binding protein 1 (TAB1), induces TAK1 autophosphorylation and activation, leading to NF-κB activation and cytokine production[38]. Recently, it was shown that protein O-GlcNAcylation is essential for T-cell development, as ablation of Ogt in T cells in vivo leads to a block of T-cell development in the thymus[8]. Furthermore, protein O-GlcNAcylation is also important for the clonal expansion of mature T cells, particularly T-cell receptor (TCR)-primed CD8$^+$ T cells[8]. Whether T and B cells employ parallel O-GlcNAcylated proteins in TCR- or BCR-mediated activation, respectively, remains unknown. It is plausible that Src family members in TCR signaling could also be O-GlcNAcylated and transduce critical signals for T-cell activation and proliferation. Given that TLR-mediated B-cell activation and proliferation is not affected in B cells lacking Ogt, we suspected that the effect of protein O-GlcNAcylation on B-cell activation may act in a pathway-dependent manner. Nevertheless, combining our data and previous findings in the literature, it appears that both TCR- and BCR-mediated activation could promote changes in protein O-GlcNAcylation, which is required for lymphocyte activation.

In addition to being crucial for B-cell activation after BCR ligation with antigen, protein O-GlcNAcylation is also involved in BAFFR-mediated mature B-cell survival. However, the development of both FO and MZ B cells has been shown to highly rely on BAFF and BAFFR[11, 12], which is partially inconsistent with the observed normal development of MZ B cells in B-KO mice. We suspect that lack of O-GlcNAcylation in other pathways may compensate for defects of BAFF signaling in MZ B cells in B-KO mice. Nevertheless, we found that several pathways downstream of ligation of BAFF and BAFFR, including the ERK5, PI3K/AKT, canonical and non-canonical NF-κB pathways, are all affected in the absence of Ogt in mature B cells. Interestingly, mice lacking Erk5 specifically in B cells also exhibited reduced number of mature B cells in bone marrow and spleen, like B-KO mice[26]. Why does a lack of Ogt selectively affect mature B-cell homeostasis? We think it is unlikely to be a result of the higher expression levels of Ogt or Oga in mature B cells because comparable levels of Ogt and Oga mRNAs were detected in various stages of development of bone marrow B cells (Supplementary Fig. 6). Identification of the important O-GlcNAcylated proteins in BAFFR-mediated signaling is awaited. It is possible that proteins that differentially lost O-GlcNAc modification in the B-KO splenic B cells identified here may also regulate the BAFFR signalosome. The Src family kinase Lyn is suggested to bridge BAFFR and BCR signaling in maintaining mature B-cell survival[10]. We found that O-GlcNAcylation of Lyn affects its interaction with Syk, and thus may be one possible underlying mechanism.

O-GlcNAcylation elevation by treatment with an OGA inhibitor, PUGNAc, or overexpression of OGT enhances and accelerates lymphocyte activation, which may be due to the increased O-GlcNAcylation of NF-ATc1 and NF-κB that causes their translocation to the nucleus[17]. Similarly, we previously demonstrated that O-GlcNAc accumulation after treatment with another OGA inhibitor, thiamet-G, enhances B-cell activation[18]. Consistently, using B-cell-specific Ogt knockout B cells, we demonstrate that O-GlcNAcylation is required for B-cell activation. Moreover, in this study, we further reveal the O-GlcNAcylated proteins that are important for the regulation of B-cell activation. We show that O-GlcNAcylation of Lyn at S19 is important for transducing signals required for B-cell activation. Phosphorylation of Lyn at S19 was present in activated macrophages[34]. However, the function of phosphorylation of Lyn at S19 and whether phosphorylation of Lyn at S19 has a competitive interplay with O-GlcNAcylation at the same site has not been demonstrated. Although our results demonstrate that SE Lyn interacts poorly with Syk and shows reduced activation after BCR cross-linking as compared with WT Lyn, we cannot directly answer whether there is a competitive interplay between O-GlcNAcylation and phosphorylation on S19 of Lyn. Interestingly, S19 lies adjacent to a conserved and functionally important caspase cleavage site, aspartic acid 18 (D18), in Lyn, which is involved in Fas- and anti-IgM-triggered apoptosis[39, 40]. Notably, the cleaved form of Lyn inhibits BCR-triggered apoptosis[40]. It is possible that O-GlcNAc modification of S19 may affect caspase-

mediated cleavage at D18, consequently modulating the outcome of apoptosis in B cells. S19 is not conserved in human Lyn; it is thus worthwhile to identify O-GlcNAc sites on human Lyn to further elucidate the underlying mechanisms of the functional effects of O-GlcNAcylation in human Lyn. Nevertheless, our result suggests that O-GlcNAcylation of Lyn is critical for the interaction with Syk and activation of Lyn in BCR activation, which may contribute to the homeostasis and activation of B cells in mice and perhaps other organisms. In addition to Lyn, we identified several other putative O-GlcNAcylated molecules in B cells. Many of these proteins are either within the BCR signalosome or important for B-cell function. The identification of the O-GlcNAcylated sites on these proteins and their relevance to B-cell activation may further advance our understanding of the functions of protein O-GlcNAcylation in B-cell activation.

Using Cγ1-KO mice, we could show that GC and memory B cells have increased levels of apoptosis, which is linked with the reduced production of antigen-specific antibody and long-lived plasma cells. The mechanisms causing the impaired formation of GC and maintenance of the survival of memory B cells as well as long-lived plasma cells in Cγ1-KO mice could also be defective BAFF and its receptor-mediated signaling pathways because BAFFR is also abundantly expressed by memory B cells and the expression of two other receptors for BAFF, cyclophilin ligand interactor (TACI) and B-cell maturation antigen (BCMA), are induced following the differentiation of plasma cells[41].

Altogether, we demonstrate a role for Ogt and protein O-GlcNAcylation in the maintenance of the homeostasis of mature B cells, memory B cells, and plasma cells. O-GlcNAcylation at S19 of Lyn is also crucial for provoking the activation of B cells after cross-linking of the BCR. Given that expression of X-linked genes, including OGT, is increased in women with lupus[42], our finding may be relevant to the design of new strategies for the control of autoimmune diseases.

## Methods

**Mice.** $Ogt^{f/f}$ and $Ogt^{f/}Y$ [2] mice were purchased from The Jackson Laboratory and crossed to $CD19^{Cre/+}$ or $IgHC\gamma1^{Cre/+}$ transgenic mice, both from The Jackson Laboratory. All mice were on a C57BL/6 background. Mice were maintained in specific pathogen-free condition in the animal facility of Institute of Cellular and Organismic Biology at Academia Sinica. All animal procedures were approved by the Institutional Animal Care and Utilization Committee of Academia Sinica. Only 6- to 8-week-old male mice were used in experiments. Due to the limitation of the availability of knockout mice and their littermate control mice, no randomization was used. Mouse experiments were not blinded. Sample size for animal experiments was decided based on prior data generated in the laboratory and guidelines for "3 Rs" of animal research. No mouse was excluded from the analyses. Primer sequences for semi-quantitative PCR analysis of Ogt genomic DNA are: 5′-CATC TCTCCAGCCCCACAAACTG-3′ and 5′-GACGAAGCAGGAGGGGAGAGCA C-3′; Il-2: 5′-CTAGGCCACAGAATTGAAAGATCT-3′ and 5′-GTAGGTGGAAA TTCTAGCATCATCC-3′; Selp: 5′-TTGTAAATCAGAAGGAAGTGG-3′ and 5′-C GAGTTACTCTTGATGTAGATCTCC-3′.

**Cell culture.** 293 T (ATCC CRL-3216) cells were maintained in DMEM (Life Technologies) containing 10% FBS, 100 U ml$^{-1}$ penicillin and 100 μg ml$^{-1}$ streptomycin at 37 °C with 5% CO$_2$. Human Ramos B cells were purchased from Bioresource Collection and Research Center, Taiwan (BCRC) (BC-60252) and were cultured in RPMI 1640 medium (Life Technologies) containing 10% FBS, 100 U ml$^{-1}$ penicillin and 100 μg ml$^{-1}$ streptomycin (Life Technologies) at 37 °C with 5% CO$_2$. All the cell lines used in this study were confirmed to be free from mycoplasma contamination by using EZ-PCR mycoplasma test kit (Biological Industries). Mouse bone marrow B cells or splenic B cells purified from 6- to 8-week-old mice were isolated by anti-B220 magnetic beads (Miltenyi Biotec) and cultured in RPMI 1640 medium with 10% charcoal/dextran-treated FBS, 100 U ml$^{-1}$ penicillin, 100 μg ml$^{-1}$ streptomycin, and 50 μM 2-ME at 37 °C with 5% CO$_2$. Recombinant BAFF (rBAFF, 200 ng ml$^{-1}$) (Peprotech) was added to bone marrow or splenic B cells ($2 \times 10^6$ cells per ml) for 72 h. Mouse splenic B cells ($2 \times 10^6$ cells per ml) were stimulated with 10 μg ml$^{-1}$ goat-anti mouse IgM F(ab′)₂ (Jackson ImmunoResearch Laboratories), 2 μg ml$^{-1}$ LPS (Sigma), or 1 μg ml$^{-1}$ anti-mouse CD40 (BD Biosciences), 25 μg ml$^{-1}$ goat-anti-mouse IgM (μ chain specific, Jackson ImmunoResearch Laboratories) plus 200 ng ml$^{-1}$ recombinant mouse IL-21 (Affymetrix eBioscience). For BrdU labeling, 10 μM BrdU (BD Biosciences)

was added into stimulated splenic B cells for 1 h before collecting the cells for flow cytometric analysis. Ramos B cells were stimulated by the treatment with goat anti-human IgM F(ab′)₂ (25 μg ml$^{-1}$) (Jackson ImmunoResearch Laboratories) for 5 min.

**Flow cytometry.** Cells from spleen and bone marrow were isolated and stained as follows[43]: single-cell suspensions, prepared by 20 μg ml$^{-1}$ Liberase Blendzyme II (Roche Applied Science) treatment for 20 min at room temperature, were washed with RPMI 1640 (Invitrogen) containing 50 μg ml$^{-1}$ DNase I (Roche Applied Science) and 3% FBS, followed by removal of RBCs by RBC lysis buffer (Sigma-Aldrich). Bone marrow cells were flushed from tibias and femurs with medium with a syringe equipped with 26 G guage needle and filtered through cell strainer (BD Falcon). Fluorescence intensity was analyzed by FACScanto (BD Biosciences) and results were analyzed using FCS Express 3.0 software. In some experiments, B cells were sorted by FACS Aria (BD Biosciences). Primary antibodies used in this study are: APC-Cy7 (allophycocyanin-Cy7)-conjugated anti-mouse B220 (clone RA3-6B2, BD PharMingen, 1:200), PE (phycoerythrin)-conjugated anti-mouse B220 (clone RA3-6B2, BD PharMingen, 1:200), biotin-conjugated anti-mouse BP1 (clone 6C3, BD PharMingen, 1:100), PerCP-Cy5 (peridinin-chlorophyll proteins-Cy5)-conjugated anti-mouse CD24 (clone M1/69, BD PharMingen, 1:100), APC (allophycocyanin)-conjugated anti-mouse CD43 (clone S7, BD PharMingen, 1:100), FITC (fluorescein isothiocyanate)-conjugated anti-mouse CD21 (clone 7G6, BD PharMingen, 1:100), PE-Cy7 (phycoerythrin-Cy7)-conjugated anti-mouse CD23 (clone B3B4, BD PharMingen, 1:100), PE-Cy7-conjugated anti-mouse IgG1 (clone RMG1-1, Biolegend, 1:200), FITC-conjugated anti-mouse GL7 (clone GL7, BD PharMingen, 1:100), PerCP-Cy5-conjugated anti-mouse CD38 (clone 90, BD PharMingen, 1:100), APC-conjugated anti-mouse CD3 (clone 145-2C11, BD PharMingen, 1:200), FITC-conjugated anti-mouse IgD (clone 11–26 c.2a, BD PharMingen, 1:100), PE-conjugated anti-mouse IgM (clone RMM-1, Biolegend, 1:100), PE-Cy7-conjugated SAV (streptavidin) (BD PharMingen, #557598, 1:200), PE-Cy7-conjugated anti-mouse CD93 (clone AA4.1, Affymetrix eBioscience, 1:100), FITC-conjugated anti-mouse CD69 (clone H1.2F3, BD PharMingen, 1:200), PE-conjugated anti-mouse CD86 (clone GL1, BD PharMingen, 1:100) and APC-conjugated anti-BrdU (clone 3D4, BD PharMingen, 1:200). For monitoring the status of cell apoptosis, Annexin-V conjugated with APC and 7-AAD staining (BD PharMingen, 1:200) were used following manufacturer's protocols. FACS gating strategies are present in Supplementary Figs. 12–15.

**Immunoprecipitation and immunoblotting.** Protein lysate preparation, immunoprecipitation, and immunoblotting were performed as follows[18]: mouse splenic B cells and Ramos B cells were collected and lysed in IP lysis buffer containing 50 mM Tris-Cl (pH 8.0), 150 mM NaCl, 5 mM EDTA, 0.5% Triton X-100, 0.1% sodium deoxycholate, protease and phosphatase inhibitor cocktail (Roche) and 10 nM OGA inhibitor, Thiamet G (Cayman Chemical). Lysates were incubated with rotation at 4 °C for 30 min and then centrifuged at 12000 rpm at 4 °C for 15 min, followed by collection of supernatant for further pre-clear by protein A beads (Santa Cruz). For immunoprecipitation, 5 μg of anti-Syk antibody (Abcam), anti-Flag antibody (Sigma) or rabbit IgG (Bethyl) were incubated with 500 μg pre-cleared cell lysates at 4 °C for overnight and then protein A agarose beads (Santa Cruz) were added to incubate for another 2 h to precipitate the protein–antibody complex. In one experiment, membrane fraction was extracted by ProteoExtract™ Native Membrane Protein Extraction kit (CALBIOCHEM) and was used for subsequent immunoprecipitation. For the purpose of checking protein O-GlcNAcylation, 50 μl succinylated wheat germ agglutinin (sWGA) agarose (Vector Laboratories) were incubated with cell lysates (250 μg) at 4 °C for 12 h, and GlcNAc (0.5 M, Sigma) was added in the control reaction. Proteins pulled down by sWGA agarose were then eluted by 5 × sample buffer containing 250 mM Tris-Cl (pH 6.8), 4% bromophenol blue, 2% SDS, 50% glycerol and 10% β-mercaptoethanol at 95 °C for 5 min for subsequent immunoblotting analysis.

pEGFP-C1 vector expressing full-length mouse Lyn fused with Flag or Flag and EGFP was constructed. The Lyn expression vector was then transfected to 293T cells by polyethylenimine (PEI) method and 48 h later, Flag-Lyn or Flag-EGFP-Lyn protein was purified by anti-Flag antibody, using as follows[18]: cells were lysed in IP lysis buffer containing 50 mM Tris-Cl (pH 8.0), 150 mM NaCl, 5 mM EDTA, 0.5% TritonX-100, 0.1% sodium deoxycholate, protease and phosphatase inhibitor cocktail (Roche) and 10 nM Thiamet G, followed by incubation on ice for 30 min. The supernatant was collected by centrifugation at 12,000 r.p.m. at 4 °C for 15 min, followed by further pre-clear with protein A beads (Santa Cruz Biotech, sc-2001). Five μg of anti-Flag antibody (Sigma, F3165) was added to the pre-cleared lysate and incubated at 4 °C for overnight, followed by incubation with protein A beads. Details for the generation of Lyn SA and SE mutants by site-directed mutagenesis will be available upon request. In one experiment, control vector, Flag-EGFP-WT, Flag-EGFP-SA or Flag-EGFP-SE Lyn was transfected into Ramos B cells by Lonza Amaxa Nucleofector V kit. Twelve hours later, lysates were prepared for immunoprecipitation by anti-Flag antibody.

Primary antibodies used in this study are anti-O-GlcNAc (clone: CTD110.6, Sigma, O77764, 1:1000), anti-phospho-Syk (Tyr525/526) (Cell Signaling, #2710, 1:500), anti-Syk (Cell Signaling, #2712, 1:500 or Abcam, ab53777, 1:500), anti-phospho-Lyn Tyr396 (Abcam, ab40660, 1:1000), anti-Lyn (Biolegend, 628102, 1:1000), anti-phospho-PI3K (p85 Tyr458/p55 Tyr199) (Cell Signaling, #4228,

1:1000), anti-PI3K p85 (Cell Signaling, #4228, 1:1000), anti-phospho-AKT (Ser473) (Cell Signaling, #9271, 1:1000), anti-AKT (Cell Signaling, #9272, 1:1000), anti-phospho-CD79a (Tyr182) (Cell Signaling, #5173, 1:1000), anti-CD79a (Abcam, ab3121, 1:1000), anti-Bim (Millipore, AB1770S, 1:1000), anti-Erk5 (Cell Signaling, #3372, 1:1000), anti-calnexin (Santa-Cruz, sc-6465, 1:1000), anti-OGT (Genetex, GTX30868, 1:1000), anti-actin (Sigma, A2228, 1:5000), anti-tubulin (Thermo, MS-581, 1:5000), anti-Flag (Sigma, F3165, 1:1000), and anti-myc (GenScript, A00172, 1:1000). Secondary antibodies used in immunoblotting are goat anti-mouse IgG horseradish peroxidase (HRP)-conjugated antibody (Sigma, A2554, 1:5000), goat anti-rabbit IgG HRP-conjugated antibody (Sigma, A0545, 1:5000) and rabbit anti-goat IgG HRP-conjugated antibody (Sigma, A5420, 1:5000). Goat anti-mouse IgG light-chain-specific antibody conjugated with HRP (Millipore, AP200P, 1:5000) was used to prevent the interference of heavy chain signal after immunoprecipitation. Western Bright Sirius chemiluminescent detection reagent (Advansta, R-030027-C50) was used to detect the immunoreactive protein signals. Chemiluminescent signal images were captured by using LAS 3000 system (Fujifilm). All the immunoblotting experiments were performed at least twice and the representative results were shown. Full-length uncropped blots are presented in Supplementary Figs. 7–11.

**RT-qPCR**. Total RNA was isolated by using RNeasy mini kit (Qiagen). The collected RNAs were reverse-transcribed to first-strand cDNA by High-Capacity cDNA Reverse Transcriptase kit (Life Technology) according to the manufacturer's suggestions. RT-qPCR analysis was subsequently performed by using Power SYBR Green PCR Master Mix (Life Technology) with the ABI prism 7300 sequence detection system (Life Technology). Primer sequences for RT-qPCR analysis of Ogt mRNA are: 5′-GCACGCATTTTTGACAGAGCT-3′ and 5′-GGCTCAAACTTAA GGCACGAA-3′, Oga mRNA are: 5′-GCAAGAGTTTGGTGTGCCTCATC-3′ and 5′- GTGCTGCAACTAAAGGAGTCCC-3′, and β-actin mRNA are: 5′-GCTGTA TTCCCCTCCATCGTG-3′ and 5′-CACGGTTGGCCTTAGGGTTCAG-3′.

**Nano LC-MS/MS analysis**. To investigate O-GlcNAc-ome of B cells, 450 μg lysates prepared from mouse splenic B cells of Ctrl and B-KO mice were collected and pulled down by 80 μl of sWGA agarose at 4 °C for overnight using lysis buffer and elution buffer noted in methods for Immunoprecipitation and immunoblotting. To identify mouse Lyn O-GlcNAc site, Flag-tagged mouse Lyn protein, expressed and purified from overexpressed 293T cells as noted in Methods section for Immunoprecipitation and immunoblotting, was subjected to SDS-PAGE, followed by in gel digestion for following mass spectrometric analysis.

The MS data were acquired on an Orbitrap Fusion mass spectrometer (Thermo Fisher Scientific) equipped with Ultimate 3000 RSLC system from Dionex (Dionex Corporation) and nanoelectrospray ion source (New Objective, Inc.). Samples in 0.1% formic acid were injected onto a self-packed precolumn (150 μm I.D. × 30 mm, 5 μm, 200 Å) at the flow rate of 10 μl min⁻¹. Chromatographic separation was performed on a self-packed reversed phase C18 nano-column (75 μm I.D. × 200 mm, 3 μm, 100 Å) using 0.1% formic acid in water as mobile phase A and 0.1% formic acid in 80% acetonitrile as mobile phase B operated at the flow rate of 300 nl min⁻¹. The MS/MS were run in top speed mode with 3 s cycles; while the dynamic exclusion duration was set to 60 s with a 25 ppm tolerance around the selected precursor and its isotopes. Monoisotopic precursor ion selection was enabled and 1 + charge state ions were rejected for MS/MS. The MS/MS analyses were carried out with the collision induced dissociation (CID) mode. Full MS survey scans from m/z 300 to 1600 were acquired at a resolution of 120,000 using EASY-IC as lock mass for internal calibration. The scan range of MS/MS spectra based on CID fragmentation method were from m/z 150–2000. The precursor ion isolation was performed with mass selecting quadrupole, and the isolation window was set to m/z 2.0. The automatic gain control (AGC) target values for full MS survey and MS/MS scans were set to 2e6 and 5e4, respectively. The maximum injection time for all spectra acquisition was 200 ms. For CID, the normalization collision energy (NCE) was set to 30%.

For data analysis, all MS/MS spectra were converted as mgf format from experiment RAW file by msConvert then analyzed by Mascot for MS/MS ion search. The search parameters included the error tolerance of precursor ions and the MS/MS fragment ions in spectra were 10 ppm and 0.6 Da and the enzyme was assigned to be trypsin with the miss cleavage number three. The variable PTMs in search parameter were assigned to include the oxidation of methionine and carbamidomethylation of cysteine. To delineate the role of O-GlcNAcylated proteins in BCR signaling, the sWGA pull-down proteins identified as <5% false discovery rate (FDR) with the ion score cut-off value 24 from Mascot search were analyzed by Ingenuity Pathway Analysis (IPA) to construct the O-GlcNAc-ome of B cells.

**BrdU labeling in mice**. Ctrl and B-KO mice were treated with BrdU (2 mg, BD Biosciences, San Jose, CA, USA) by intraperitoneal injection and euthanized 12 h later to examine the rate of BrdU incorporation in bone marrow mature B cells.

**Mouse immunization and antibody analyses**. Mice were immunized with 100 μg NP₃₂-KLH (Biosearch Technologies, Petaluma, CA, USA) via intraperitoneal injection. Four weeks after primary immunization, mice were injected with 100 μg

NP₃₂-KLH for the secondary immunization. In one experiment, bone marrow cells (1 × 10⁶ cells) isolated from Ctrl and Cγ1-KO mice at 4 weeks after the secondary immunization with NP-KLH were subjected to ELISPOT analysis to test the number of NP-specific IgG-secreting plasma cells as follows[44]: B cells were seeded into a 96-well plate and incubated on nitrocellulose membranes (Bio-Rad), which was coated with NP₃₂-BSA (100 μg ml⁻¹, Biosearch technologies), at 37 °C for 2.5 h. Attached cells were lysed by PBS-EDTA for 15 min, followed by two washes with 0.1% Tween 20 in PBS and one wash in PBS. The membrane was incubated at room temperature for 30 min with PBS and 1% bovine serum albumin (BSA), followed by incubation with anti-mouse IgG (Southern Biotechnology) at a dilution of 1000 × in PBS plus 1% BSA at room temperature for 1 h. NP-specific IgG-secreting cells were then visualized by the addition of enzyme substrate. Photomicrographs of the spots were analyzed with an AID EliSpot Reader (AID Auto-immun Diagnostika GmbH, Strassberg, Germany). Sera collected at various days after immunization were subjected to ELISA analysis as follows[44]: serially diluted sera were added onto NP₃₂-BSA (100 μg ml⁻¹, Biosearch technologies) coated plates at 4 °C for overnight, followed by detection with 5 μg ml⁻¹, 156.25 ng ml⁻¹, 2.5 μg ml⁻¹ and 1.25 μg ml⁻¹ horseradish-peroxidase-conjugated anti-mouse-IgG1, IgG2b, IgG2c, and IgG3 (Bethyl Laboratories), respectively. The absorbance at 450 nm was measured using a microplate reader (SpectraMax M2, Molecular Device, Sunnyvale, CA, USA).

**Immunohistochemical and TUNEL staining**. Two weeks after immunization, isolated mouse spleens were fixed by 10% formaldehyde and embedded in paraffin. Sections, sliced at 3 μm thickness, were dewaxed and incubated with retrieval solution (pH 9; DakoCytomation) with heating (95 °C for 20 min). After blocking with 3% human serum (Invitrogen), slides were incubated with biotinylated PNA (1:100; Vector Laboratories) at 4 °C for overnight and then immersed in streptavidin-alkaline phosphatase (1:500; Vector Laboratories) followed by color development with NBT/BCIP substrates (Roche). After washing with tap water, slides were incubated with anti-B220 antibody (1:200; Abcam) at 4 °C for overnight. Immunoreactive signals were visualized by incubating with anti-rat IgG conjugated with HRP (Sigma) followed by color development with DAB (DAKO). For quantification of GC B cells, spleen sections were analyzed. In each section, five GCs were randomly selected for counting PNA⁺ cells in fixed area (400 × 400 μm).

For TUNEL staining, mice were killed 2 weeks after immunization. Spleens were collected and fixed in 10% formaldehyde (Mallinckrodt Chemicals) at 4 °C, and further processed for paraffin embedding. Samples sections were prepared at a thickness of 3 μm. Subsequent TUNEL staining was performed with an in situ cell death detection kit, Fluorescein (Roche) according to the manufacturer's suggestions. Images were acquired by Leica DMG000B microscope equipped with a 20 × /0.7 HC-Plan APO objective and Leica DFC490 camera and were analyzed by Leica FW4000 software (Leica Microsystems).

**Statistical analysis**. Our data mostly meet the assumptions of the tests (normal distribution). Statistical significance was analyzed by two-tailed unpaired t tests, in which the variances of the groups are similar. A value of P < 0.05 was considered statistically significant. All the data shown in this study are the mean ± SEM from at least three biological replicates.

**Data availability**. Mass spectrometry proteomics data that support the findings of this study have been deposited in the ProteomeXchange Consortium via the PRIDE partner repository with the accession code PXD007050. The authors declare that all other data are available from the corresponding authors upon request.

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

## Acknowledgements

The authors thank Penk-Yeir Low, Hsin-Yu Chen, Shii-Yi Yang and Yi-Ting Hsieh for technological assistance. This work was supported by Academia Sinica (AS-105-TP-A05-3) and by the Ministry of Science and Technology (106-2320-B-001-011-MY3 and 105-0210-01-13-01 to K.-I.L. and 104-2311-B-001-017-MY3 to T.A.), Taiwan.

## Author contributions

J.-L.W., M.-F.C., T.A. and K.-I.L. designed the research; J.-L.W., M.-F.C., P.-H.H., D.-Y.T., K.-H.H. and Y.-H.W. performed the experiments; J.-L.W., M.-F.C., P.-H.H., T.A. and K.-I.L. analyzed the results; T.A. and K.-I.L. wrote the paper.

## Additional information

**Competing interests:** The authors declare no competing financial interests.

