## [Peer Review File · Nature Communications]

Reviewers' comments:

Reviewer #1 (Remarks to the Author):

The paper by Wu et al. describes the loss of an intracellular carbohydrate modification, the O-GlcNAcylation, in B cells and demonstrates interesting consequences of this loss on BCR signalling, on apoptosis of B cells and on antibody responses. The study is overall well done and shows interesting results. All experiments are performed very thoroughly. The findings are novel and of general interest. Some points still have to be addressed.

1. The effect of the mutation on apoptosis is an important point in the manuscript. Examples of FACS stainings with Annexin V are shown, but for these experiments statistical analyses are not done. Show quantification of several experiments with statistics in Fig.2 and Fig.3A. Fig.2C is an unusual representation of the data. Positive values for the amount of apoptosis in both mice would be much clearer.

2. Suppl. Fig. 3 shows the O-GlcNAc-ome of activated B cells. The IPA map summarizes many found signalling proteins. Since this is a pulldown assay with a lectin it is unclear how specific these interactions are. This should be done in parallel with WT and KO B cells and the background from the KO B cells should be subtracted from the WT values. Otherwise the specificity of the pull down is unclear.

3. The authors use the Cgamma1-Cre mouse to delete the gene late after B cell activation. The authors write on page 13, line 220, that this Cre mouse line deletes just in class-switched B cells. This is not true, as ref. 32 shows. This Cre line does not delete only in IgG1 class switched B cells, but quite generally in GC B cells. This is probably due to an early induction of germline transcripts in the GC. Therefore the authors should also look at apoptosis induction in total GC B cells (Fig.5b) and report on the switching to other Ig isotypes (Fig.5e).

Reviewer #2 (Remarks to the Author):

In this manuscript the authors use various conditional OGT knock out mouse lines to that without OGT there is a loss of mature B cells, defects in BCR signaling and defects in response to BAFF. Additionally, they identified a novel O-GlcNAc site on Lyn which influences the interaction with Syk.

The authors also made Use of a IghCy1-Cre mice to create a mouse line in which the Ogt allele was presumably deleted only in class-switched B cells. Interestingly post GC B cells also fail to evoke an antibody response.

Several problems emerge with the manuscript:

While the induction of the apoptotic pathways is a clear strength of the manuscript, the linkage to the identified targets is tenuous. Certainly, changes in apoptotic targets are documented and may prove interesting. However, with potentially thousands of targets, these may not be the driving the phenotype. In addition, mosaicism in knockout is not well addressed throughout.

Figure 2C) Showing the data as a percentage of reduced apoptosis is a confusing way to display the data. The authors should display the data more clearly. Maybe by graphing each data point as percentage of apoptotic cells before and after rBAFF.

Figure 3E and 5A) the axis are not labeled with numbers.

In summary, Cre-mediated loss of OGT is a blunt tool to ask questions about the potential role of OGT in maintaining homeostasis. With so many potential targets, the targets identified may not be the critical ones.

Reviewer #3 (Remarks to the Author):

Overall interesting set of observations describing the consequences of loss of O-GlcNAcylation. There are relatively dramatic effects on B cell homeostasis and on BCR signaling in particular. Curiously and interestingly, there is a major effect on Lyn tyrosine kinase, although the exact relationship between these changes in Lyn and the B cell phenotype are not fully explored in this report.

There remain, however, a number of issues that require attention before the work is suitable for publication in my opinion.

Figure 2a: Percent annexin V positive among B220+CD43⁻ in KO is misleading as this fraction in the KO lacks mature, recirculating cells so is enriched for pre-B and immature B that will be undergoing more apoptosis.

Figure 2b: Same criticism. OK to have increased proportion but if there is some fraction of immature that are counter-selected on entry into mature, that number will be the same in both groups but appear as a much larger proportion in the KO, as there are half the number of mature B cells.

If they wish to make conclusions on lifespan, they should do traditional experiments of measuring population turnover by BrdU for example.

Figure 3 needs quantification of westerns and indication of repeats of experiments.

Figure 4 needs quantification of blot intensity and number of repeats for each experiment and blot.

Figure 5 is very poorly annotated. No indication of number of times experiment in 5a was done or any quantification. Same for number of repeats in 5b. No indication of number of sections or independent mice or experiments used in 5c. How was 5c quantification done?

I realise methods say all data are the means +/- SEM of at least three biological repeats, but there are several cases of no errors or quantification and this is particularly true for biochemistry. Also, why SEM and not SD? At the moment, for example, it looks like Fig 5 is based on 4 mice!

Reviewer #4 (Remarks to the Author):

The manuscript under review is Wu et al., "Mature B cell homeostasis, B cell activation and antibody response require protein O-GlcNAcylation," Nature Communications manuscript number NCOMMS-17-03066. The authors investigate the role of O-linked β -N-acetylglucosamine (O-GlcNAc), a widespread and essential post-translational modification, in murine B cell development and function. The authors use conditional knockout of O-GlcNAc transferase (OGT) in the hematopoietic compartment to show that O-GlcNAcylation is required for mature B cell survival and function. Through a combination of immunological, cellular and molecular biology experiments, the authors provide evidence that OGT is necessary for normal functioning of both the B cell receptor (BCR) and B cell activating factor receptor (BAFFR) pathways, and that a putative glycosylation site on the Lyn tyrosine kinase may mediate all or some of the effects of OGT in these signaling pathways. It is already well established that OGT is required for the survival and function of several differentiated cell types, including lymphocytes, but the authors have extended this work into B cell biology, providing a moderate but distinct advance in understanding the physiological functions of O-GlcNAc signaling. I believe the novelty of the work is adequate for Nature Communications, but I do have several questions and critiques that I believe should be addressed by the authors before the manuscript could be acceptable for publication.

Major points

1. The authors should apply appropriate tests of statistical significance to all their quantitative assay data (e.g., flow cytometry). For example, in Figure 3a, has the experiment been performed more than once? Is the 38.4% versus 54.5% Annexin V-positive cells in wild type versus OGT knockout statistically significant? These considerations seem especially important when the authors make claims about assays that compare different stimuli to each other. For example, in Figure 3f, the authors claim that wild type versus OGT knockout cells have significant differences in proliferation in response to anti-IgM (21.5% versus 5.69%) but not LPS (54.2% versus 42.5%). However, no tests of statistical significance are applied to support this claim. It may be that the difference in the LPS experiment is also significant. A similar problem is seen in Figure 3e, where shoulders appear in the histograms of LPS-treated OGT knockout cells, despite the authors' claim that OGT loss does not affect the response to LPS.
2. I have three concerns about the interpretation and significance of the Lyn S19 O-GlcNAcylation results. First, as noted by the authors, S19 is a known phosphorylation site, and so the S19A mutant used by the authors will ablate both phosphorylation and glycosylation at that site, confounding the interpretation of several experiments (e.g., Figures 4d-f). The authors should repeat these experiments with an S19D phospho-mimetic mutant to test whether this mutant phenocopies wild type Lyn, S19A, or neither. These results will help to determine whether loss of S19 glycosylation or phosphorylation accounts for the Syk binding and signaling differences that the authors observe. Second, S19 lies adjacent to a known and functionally important caspase cleavage site in Lyn (for example, see: Luciano et al 2001 *Oncogene* 20 4935; Gamas et al 2009 *Leukemia* 23 1500; Marchetti et al 2009 *EMBO* 28 2449). It may be that S19 O-GlcNAcylation (or phosphorylation) inhibits caspase cleavage at this site, affecting Lyn signaling and downstream apoptotic phenotypes. At the very least, the authors should discuss these literature reports in their manuscript and interpret their own results in light of this information. Third, the Lyn S19 site is not conserved in humans (leucine) or even closely related rodents (methionine in rat). This lack of conservation calls into question the physiological significance of S19 O-GlcNAcylation. At a minimum, the authors should comment in the manuscript on whether they believe that Lyn glycosylation may govern human B cell development or function.
3. The description of the mass spec proteomics experiments is inadequate. The authors should include details in the Materials and Methods about the sample preparation, including buffer composition, washing and elution conditions, etc. Furthermore, the supplementary Excel file provides far too little detail on the proteomics results. The authors should provide separate protein ID lists for each independent experiment, and should provide standard information such as peptide number and percent sequence coverage for each protein ID, false discovery rate and minimum peptide number cut-offs used in the analysis, etc.

Minor points

1. I suggest moving Supplementary Figure 2a into Figure 1, to facilitate the reader's understanding of the authors' experimental strategy.
2. I'm not sure I understand the authors' interpretation of Figure 2c. The authors seem to conclude that OGT is required for normal BAFFR signaling, and yet the exogenously added BAFF still suppresses apoptosis in OGT knockout cells (albeit less well than in wild type cells). If OGT is required for BAFF signaling, then why does exogenous BAFF still have an effect on OGT knockout cells? Can the authors clarify their interpretation in the text?
3. I believe there is a typo in Supplementary Figure 3c, where "anti-IgM" is meant, not "0.5 M GlcNAc."
4. Scale marks and numbers are missing on the x- and y-axes of the plots in Figures 3e and 5a.

5. In revising, the authors should take care to fix the errors of standard English usage throughout the manuscript.

To Reviewer #1

The paper by Wu et al. describes the loss of an intracellular carbohydrate modification, the O-GlcNAcylation, in B cells and demonstrates interesting consequences of this loss on BCR signalling, on apoptosis of B cells and on antibody responses. The study is overall well done and shows interesting results. All experiments are performed very thoroughly. The findings are novel and of general interest. Some points still have to be addressed.

We thank the reviewer for the positive comments and for acknowledging the novelty of our manuscript. We are also grateful for the reviewer's constructive suggestions. Our point-by-point responses are listed below.

1. The effect of the mutation on apoptosis is an important point in the manuscript. Examples of FACS stainings with Annexin V are shown, but for these experiments statistical analyses are not done. Show quantification of several experiments with statistics in Fig.2 and Fig.3A. Fig.2C is an unusual representation of the data. Positive values for the amount of apoptosis in both mice would be much clearer.

Reply:

Thank you for the thoughtful suggestions. We added the quantitative results of three independent experiments associated with Figures 2a, 2b and 3a.

With regard to the "unusual representation of data" in Fig. 2c (Fig. 2d in this revision), due to variable apoptotic rates in biological replications, more consistent and reproducible results are obtained when we convert the results to "% of reduction of apoptosis", which is calculated by the following formula: $(\% \text{ of apoptosis in untreated group} - \% \text{ of apoptosis in rBAFF treated group}) / (\% \text{ of apoptosis in untreated group})$. We have redone the experiments by analyzing the frequency of Annexin V⁺ cells in bone marrow B and splenic B cells treated with rBAFF for 72 hrs, instead of 16 hrs in the original version. Our new data showed that bone marrow and splenic B cells from B-KO mice are protected less by rBAFF treatment (Fig. 2d in this revision).

2. Suppl. Fig. 3 shows the O-GlcNAc-ome of activated B cells. The IPA map summarizes many found signalling proteins. Since this is a pulldown assay with a lectin it is unclear how specific these interactions are. This should be done in parallel with WT and KO B cells and the background from the KO B cells should be subtracted from the WT values. Otherwise the specificity of the pull down is unclear.

Reply:

We thank reviewer for the suggestion. According to the reviewer's suggestion, we have re-performed the sWGA pull-down and mass spectrometric analysis by using lysates prepared from Ctrl and B-KO splenic B cells (Supplementary Fig. 4a in this revision and Supplementary Table 1). IPA analysis of the differentially O-GlcNAcylated proteins found

only in Ctrl B cells revealed again that proteins in BCR signaling pathway are enriched (Supplementary Fig. 4b).

3. The authors use the Cgamma1-Cre mouse to delete the gene late after B cell activation. The authors write on page 13, line 220, that this Cre mouse line deletes just in class-switched B cells. This is not true, as ref. 32 shows. This Cre line does not delete only in IgG1 class switched B cells, but quite generally in GC B cells. This is probably due to an early induction of germline transcripts in the GC. Therefore the authors should also look at apoptosis induction in total GC B cells (Fig.5b) and report on the switching to other Ig isotypes (Fig.5e).

Reply:

We thank the reviewer for pointing out our imprecise statement and the excellent suggestions. We have corrected the sentence related to the activity of Cre in C_γ1-KO mice (page 14). We also examined the proportion of Annexin V⁺ GC B cells and the levels of NP-specific IgG_{2b}, IgG_{2c} and IgG₃, which were detectable in sera using our immunization protocols, after secondary NP-KLH immunization. Our new results are shown in Fig. 5b and Supplementary Fig. 5c in this revision. We found that C_γ1-KO GC B cells are more apoptotic, and that the titers of NP-specific IgG_{2b}, IgG_{2c} and IgG₃ are all reduced in C_γ1-KO mice.

To Reviewer #2

In this manuscript the authors use various conditional OGT knock out mouse lines to that without OGT there is a loss of mature B cells, defects in BCR signaling and defects in response to BAFF. Additionally, they identified a novel O-GlcNAc site on Lyn which influences the interaction with Syk.

The authors also made Use of a IghC γ 1-Cre mice to create a mouse line in which the *Ogt* allele was presumably deleted only in class-switched B cells. Interestingly post GC B cells also fail to evoke an antibody response.

Several problems emerge with the manuscript:

Reply:

We thank the reviewer for the thorough evaluation of our manuscript. Our point-by-point responses are listed below.

While the induction of the apoptotic pathways is a clear strength of the manuscript, the linkage to the identified targets is tenuous. Certainly, changes in apoptotic targets are documented and may prove interesting. However, with potentially thousands of targets, these may not be driving the phenotype. In addition, mosaicism in knockout is not well addressed throughout.

Reply:

We thank the reviewer for raising these critical and important points. We agree that changes in apoptotic targets may provide direct connection between loss of *O*-GlcNAcylation and B cell apoptosis. However, our data also suggest that molecules closely associated with B cell receptor may be malfunctioning in B-KO B cells. BCR has been shown to be critical for maintaining tonic signaling of mature B cells, for bridging BAFF signaling as well as for mounting B cell activation. Our new *O*-GlcNAc proteome results comparing proteins pulled-down by sWGA from Ctrl and B-KO splenic B cell lysates revealed nearly 400 candidate proteins (Supplementary Table 1), and IPA analysis revealed that the proteins involved in BCR signaling are enriched among the identified *O*-GlcNAcylated proteins (Supplementary Fig. S4b). Besides, IPA analysis did not reveal any significant pathways/molecules directly associated with B cell apoptosis.

With regard to the mosaicism in Cre-mediated knockout mice, we agree that Cre-mediated recombination may not give rise to a complete deletion of the gene of interest. A previous report generating heart specific *Ogt* knockout mice (PNAS, **107**, 17797-17802, (2010)) also observed 40% residual *Ogt* mRNA and protein *O*-GlcNAcylation. We have been aware of this mosaicism issue and used only male mice in experiments as *Ogt* is X-linked, which was described on page 6.

Figure 2C) Showing the data as a percentage of reduced apoptosis is a confusing way to display the data. The authors should display the data more clearly. Maybe by graphing each

data point as percentage of apoptotic cells before and after rBAFF.

Reply:

Thank you for pointing out the potential problem in the data presentation in Figure 2c. However, due to variable apoptotic rates in biological replications, more consistent and reproducible results are obtained when we convert the results to “% of reduction of apoptosis”, which is calculated by the following formula: $(\% \text{ of apoptosis in untreated group} - \% \text{ of apoptosis in rBAFF treated group}) / (\% \text{ of apoptosis in untreated group})$. We have redone the experiments by examining the differences in apoptosis in Ctrl and B-KO B cell cultures in response to rBAFF treatment at 72 hrs. One representative flow cytometric analysis data and statistical results from three independent repeats are shown in this revision. Our new data showed that bone marrow and splenic B cells from B-KO mice are protected less by rBAFF treatment (Fig. 2d in this revision).

Figure 3E and 5A) the axis are not labeled with numbers.

Reply:

We apologize for the mistakes, which have been corrected in this revision.

In summary, Cre-mediated loss of OGT is a blunt tool to ask questions about the potential role of OGT in maintaining homeostasis. With so many potential targets, the targets identified may not be the critical ones.

Reply:

We agree that the phenotypes observed here are unlikely attributed to one particular protein. Indeed, our proteome analysis of the differentially *O*-GlcNAcylated proteins between Ctrl and B-KO splenic B cells also revealed nearly 400 possible candidates. We thus took the approach to analyze the function of these candidates by IPA that revealed the over-representation of the proteins involved in BCR pathway. We picked Lyn for further study because Lyn is an important molecule closely associated with BCR that would affect the function of BCR signaling immediately following to crosslinking of BCR.

Reports showing the physiological functions of *O*-GlcNAcylation *in vivo* have started being published only recently by using Cre-mediated ablation of *Ogt*, including “Loss of *O*-GlcNAc glycosylation in forebrain excitatory neurons induces neurodegeneration” (PNAS, 2016, 27:15120-15125), “Glucose and glutamine fuel protein *O*-GlcNAcylation to control T cell self-renewal and malignancy” (Nat Immunol., 2016, 17:712-720), and “Disruption of *O*-linked N-Acetylglucosamine Signaling Induces ER Stress and β Cell Failure” (Cell Rep. 2015, 13:2527-38)”. Although Cre-mediated loss of OGT may be a blunt tool, we believe it is an important step forward to reveal the significance of *O*-GlcNAcylation in various biological systems.

To Reviewer #3

Overall interesting set of observations describing the consequences of loss of O-GlcNAcylation. There are relatively dramatic effects on B cell homeostasis and on BCR signaling in particular. Curiously and interestingly, there is a major effect on Lyn tyrosine kinase, although the exact relationship between these changes in Lyn and the B cell phenotype are not fully explored in this report.

There remain, however, a number of issues that require attention before the work is suitable for publication in my opinion.

We thank the reviewer for considering our manuscript interesting, and for the insightful suggestions. Our point-by-point responses are listed below.

Figure 2a: Percent annexin V positive among B220⁺CD43⁻ in KO is misleading as this fraction in the KO lacks mature, recirculating cells so is enriched for pre-B and immature B that will be undergoing more apoptosis.

Figure 2b: Same criticism. OK to have increased proportion but if there is some fraction of immature that are counter-selected on entry into mature, that number will be the same in both groups but appear as a much larger proportion in the KO, as there are half the number of mature B cells.

If they wish to make conclusions on lifespan, they should do traditional experiments of measuring population turnover by BrdU for example.

Reply:

Thank you for pointing out the potential pitfall in the interpretation of data presented in Figures 2a and 2b. With regard to Figure 2a, it is possible that the B220⁺CD43⁻ population in B-KO contains higher proportion of apoptotic cells, not because mature B cells in B-KO are more apoptotic but because the population contains higher proportion of pre-B and immature B cells, which may be more apoptosis-prone. However, this possibility is addressed in Figure 2b, which shows that pre-B and immature B cells are actually less apoptotic than mature B cells; thus, the higher proportion of pre-B and immature B cells in the B220⁺CD43⁻ population of B-KO actually dilutes the effect of *Ogt* deficiency on the overall apoptosis in this population. With regard to Figure 2b, it is possible that the mechanism the reviewer proposed (if some fraction of immature B cells undergo apoptosis upon entry into mature B cells, that number will be the same in both WT and B-KO but appear as a much larger proportion in the B-KO, as there are less mature B cells in B-KO) contributes to the observed effect to some extent, but it would not ultimately explain why there are less mature B cells in B-KO mice in the first place. Nevertheless, to address the point raised by the reviewer, we performed in vivo BrdU labeling experiment (Figure 2c). Our new results showed that bone marrow mature B cells in B-KO mice incorporated much less BrdU (Figure 2c), supporting our notion that O-GlcNAcylation is required for maintaining the homeostasis of mature B cells.

Figure 3 needs quantification of westerns and indication of repeats of experiments.

Reply:

Thank you for the suggestion. The quantification results of the representative immunoblots were indicated below in Figure 3. The statistical results from three independent immunoblotting analyses were shown in Supplementary Figures 3c, d and e. Additionally, in Figures 2 e and f, quantification results of the representative immunoblots were indicated. The statistical results from three independent immunoblotting analyses were shown in Supplementary Figures 3a and b. The number of experimental repeats was indicated in the Figure Legend.

Figure 4 needs quantification of blot intensity and number of repeats for each experiment and blot.

Reply:

The quantification results of the representative immunoblots were indicated below to each blot in Figure 4, and the number of experimental repeats was indicated in the Figure Legend.

Figure 5 is very poorly annotated. No indication of number of times experiment in 5a was done or any quantification. Same for number of repeats in 5b. No indication of number of sections or independent mice or experiments used in 5c. How was 5c quantification done?

Reply:

We apologize that our annotation of Figure 5 lacked clarity. The experiments in Figure 5a have been repeated three times. The statistical analyses are provided in the right panel of Figure 5a in this revision. In Figure 5b, we performed three independent experiments, in one representative experiment shown in Figure 5b, we used 4 pairs of Ctrl and C γ 1-KO mice for NP-KLH immunization and checked the apoptosis rate in B cell subsets 14 days after immunization. Regarding to Figure 5c, two pairs of Ctrl and C γ 1-KO mice were immunized and sacrificed at 14 days after NP-KLH immunization for staining with PNA and B220 in each experiment and two biological replications have been done. To quantify GC B cells in NP-KLH immunized Ctrl and C γ 1-KO mice in Figure 5c, five GCs were randomly selected to count the number of PNA⁺B220⁺ cells in fixed area (400 \times 400 μ m), which was described in the Materials and Methods in this revision. The number of mice we used for each panel was indicated in the Figure Legends in the original version and further clarified in this revision.

I realise methods say all data are the means +/- SEM of at least three biological repeats, but there are several cases of no errors or quantification and this is particularly true for biochemistry. Also, why SEM and not SD? At the moment, for example, it looks like Fig 5 is based on 4 mice!

Reply:

In this revision, we include the results of statistical analyses along with our original data. We have also ensured that the numbers of mice or experiments have been listed in each Figure Legend. We have used more than 4 pairs of mice to perform the experiments in Figure 5. All quantitative plots in this manuscript are shown as mean \pm SEM, except in Figure 5b and 5e (which show individual values and mean). We chose to show data as mean \pm SEM, as our aim was to compare means by statistical test (rather than to show variability). Whether SD or SEM was used in the plots does not change the statistical analysis (*P* value).

To Reviewer #4

The manuscript under review is Wu et al., "Mature B cell homeostasis, B cell activation and antibody response require protein O-GlcNAcylation," Nature Communications manuscript number NCOMMS-17-03066. The authors investigate the role of O-linked β -N-acetylglucosamine (O-GlcNAc), a widespread and essential post-translational modification, in murine B cell development and function. The authors use conditional knockout of O-GlcNAc transferase (OGT) in the hematopoietic compartment to show that O-GlcNAcylation is required for mature B cell survival and function. Through a combination of immunological, cellular and molecular biology experiments, the authors provide evidence that OGT is necessary for normal functioning of both the B cell receptor (BCR) and B cell activating factor receptor (BAFFR) pathways, and that a putative glycosylation site on the Lyn tyrosine kinase may mediate all or some of the effects of OGT in these signaling pathways. It is already well established that OGT is required for the survival and function of several differentiated cell types, including lymphocytes, but the authors have extended this work into B cell biology, providing a moderate but distinct advance in understanding the physiological functions of O-GlcNAc signaling. I believe the novelty of the work is adequate for Nature Communications, but I do have several questions and critiques that I believe should be addressed by the authors before the manuscript could be acceptable for publication.

We thank the reviewer for recognizing the novelty of our manuscript. Our point-by-point responses are listed below.

Major points

1. The authors should apply appropriate tests of statistical significance to all their quantitative assay data (e.g., flow cytometry). For example, in Figure 3a, has the experiment been performed more than once? Is the 38.4% versus 54.5% Annexin V-positive cells in wild type versus OGT knockout statistically significant? These considerations seem especially important when the authors make claims about assays that compare different stimuli to each other. For example, in Figure 3f, the authors claim that wild type versus OGT knockout cells have significant differences in proliferation in response to anti-IgM (21.5% versus 5.69%) but not LPS (54.2% versus 42.5%). However, no tests of statistical significance are applied to support this claim. It may be that the difference in the LPS experiment is also significant. A similar problem is seen in Figure 3e, where shoulders appear in the histograms of LPS-treated OGT knockout cells, despite the authors' claim that OGT loss does not affect the response to LPS.

Reply:

We thank the reviewer for the thoughtful suggestions. The statistical analyses associated with the flow cytometric results in Figures 3a, e and f are included in this revision.

2. I have three concerns about the interpretation and significance of the Lyn S19

O-GlcNAcylation results. First, as noted by the authors, S19 is a known phosphorylation site, and so the S19A mutant used by the authors will ablate both phosphorylation and glycosylation at that site, confounding the interpretation of several experiments (e.g., Figures 4d-f). The authors should repeat these experiments with an S19D phospho-mimetic mutant to test whether this mutant phenocopies wild type Lyn, S19A, or neither. These results will help to determine whether loss of S19 glycosylation or phosphorylation accounts for the Syk binding and signaling differences that the authors observe. Second, S19 lies adjacent to a known and functionally important caspase cleavage site in Lyn (for example, see: Luciano et al 2001 *Oncogene* 20 4935; Gamas et al 2009 *Leukemia* 23 1500; Marchetti et al 2009 *EMBO* 28 2449). It may be that S19 O-GlcNAcylation (or phosphorylation) inhibits caspase cleavage at this site, affecting Lyn signaling and downstream apoptotic phenotypes. At the very least, the authors should discuss these literature reports in their manuscript and interpret their own results in light of this information. Third, the Lyn S19 site is not conserved in humans (leucine) or even closely related rodents (methionine in rat). This lack of conservation calls into question the physiological significance of S19 O-GlcNAcylation. At a minimum, the authors should comment in the manuscript on whether they believe that Lyn glycosylation may govern human B cell development or function.

Reply:

We thank the reviewer for these excellent comments.

(1) Phosphorylation at S19

We generated a S19E mutant Lyn in which Ser 19 was replaced by glutamic acid, to act as a phospho-mimicry mutation. The experiments in Figures 4d, e and f were re-conducted. Our new results showed that, like S19A, S19E mutant was not O-GlcNAcyated (Fig. 4d), interacted poorly with Syk (Fig. 4e) and was not activated well in response to anti-IgM crosslinking (Fig. 4f). Therefore, O-GlcNAcylation, not phosphorylation, at S19 is critical for Lyn activation and interaction with Syk. Besides, although there are at least three proteomics papers showing that Lyn is phosphorylated at S19 (*Molecular cell* (2009) **36**, 326-339, *Science signaling* (2013) **6**, rs11, and *Immunity* (2009) **30**, 143-154) as noted by the reviewer, whether S19 is phosphorylated in B cells is still unclear. Nevertheless, our data strongly imply that the functional deficiency of Lyn S19A mutant results from the loss of O-GlcNAcylation.

(2) Influence on the adjacent caspase cleavage site (D18)

This is an interesting point we were not aware of, and we thank the reviewer for pointing out this possibility. It is possible that the presence of O-GlcNAc modification on S19 site might affect the cleavage of Lyn by caspase on the adjacent D18 site, which might regulate the downstream apoptosis events. We have included this excellent point in Discussion on pages 18 in this revision. The references kindly provided by the reviewer were also cited.

(3) Non-conservation of S19

This is an insightful comment. As noted by the reviewer, S19 is not a conserved amino acid. It

is possible that other serine or threonine sites near D18 site in human Lyn, such as S11, S13 and T21, are *O*-GlcNAcylated. Given that there is no conclusive consensus motif for *O*-GlcNAcylation (Clinical proteomics (2014) **11**, 8), it is quite possible that *O*-GlcNAcylation in human Lyn might occur on another site, such as the abovementioned sites. As suggested by the reviewer, we have included this point in Discussion in this revision on pages 18-19.

3. The description of the mass spec proteomics experiments is inadequate. The authors should include details in the Materials and Methods about the sample preparation, including buffer composition, washing and elution conditions, etc. Furthermore, the supplementary Excel file provides far too little detail on the proteomics results. The authors should provide separate protein ID lists for each independent experiment, and should provide standard information such as peptide number and percent sequence coverage for each protein ID, false discovery rate and minimum peptide number cut-offs used in the analysis, etc.

Reply:

We thank the reviewer for the thoughtful suggestions. The detailed experimental procedures, including sample preparation, for mass spec proteomic analysis were cited with appropriate references or provided in the Materials and Methods in this revision. More detailed descriptions in the supplementary Excel file were also provided.

Minor points

1. I suggest moving Supplementary Figure 2a into Figure 1, to facilitate the reader's understanding of the authors' experimental strategy.

Reply:

We thank the reviewer for the suggestion. Supplementary Figure 2a has been moved to Figure 1b in this revised version.

2. I'm not sure I understand the authors' interpretation of Figure 2c. The authors seem to conclude that OGT is required for normal BAFFR signaling, and yet the exogenously added BAFF still suppresses apoptosis in OGT knockout cells (albeit less well than in wild type cells). If OGT is required for BAFF signaling, then why does exogenous BAFF still have an effect on OGT knockout cells? Can the authors clarify their interpretation in the text?

Reply:

Our results in Figure 2c (Figure 2d in this revised version) demonstrated that BAFF could not rescue the apoptosis of B-KO B cells as efficiently as that of Ctrl B cells. As the reviewer pointed out, BAFF still provide some protection to B-KO B cells. This may imply that *Ogt* is not absolutely required for BAFF signaling pathways. As the phrase "BAFF-mediated mature B cell survival requires *Ogt*" may be an over-statement, we re-phrased it as "BAFF-mediated mature B cell survival is enhanced by *Ogt*" (page 9).

3. I believe there is a typo in Supplementary Figure 3c, where “anti-IgM” is meant, not “0.5 M GlcNAc.”

Reply:

We thank the reviewer for pointing out this confusing annotation. We have corrected Supplementary Figure 3c (Figure 4c in this revision).

4. Scale marks and numbers are missing on the x- and y-axes of the plots in Figures 3e and 5a.

Reply:

Thank you very much for noting our omission. The errors have been corrected in this revision.

5. In revising, the authors should take care to fix the errors of standard English usage throughout the manuscript.

Reply:

We thank the reviewer for the suggestion. This revised manuscript has been proofread by professional English editing services by Edanz (order number 29265).

REVIEWERS' COMMENTS:

Reviewer #1 (Remarks to the Author):

No further comments. All my points have been addressed. Thank you.

Reviewer #2 (Remarks to the Author):

Manuscript is appropriately amended.

Reviewer #3 (Remarks to the Author):

The main points of the manuscript are, that in the absence of O-GlcNAcylation there is:

- Increased apoptosis of mature B cells
- Decreased signalling of B cells after BCR ligation
- OGN-Acylation of Lyn is crucial for its activation and Syk interaction
- Increased apoptosis of GC B cells, memory B cells and plasma cells

The apoptosis conclusion is interesting but I think more complex than is apparent from reading the manuscript and this is why I had asked for the BrdU labeling experiment, which has now been done. The aspects which I find puzzling are that it is restricted to the recirculating B cell compartment in the bone marrow, at least statistically, so it may not be so straightforward to say that apoptosis is the cause of the peripheral B cell deficiency. That is the reason for the BrdU experiment, which has been done, but on the bone marrow B cells. The data presented will require some explanation by the authors to put them into the context of an apoptosis-only model as they suggest a 66% reduced production of mature BM B cells in the KO within the 12 hour pulse. Wouldn't one have expected increased proportional production in the KO compartment if there was increased apoptosis but similar input? At the moment, there is both increased apoptosis and potentially decreased input, meaning that both could be responsible for the deficit in the population. Equally, there are no data turnover for the peripheral B cell populations.

Finally, and I apologize for raising this now but I don't think any of the data presented in either the first or revised submission address this point, I think the abstract has to be modified to remove the claim of increased apoptosis in plasma cells in the KO. No evidence was presented on this, I don't think, and at the moment, it is hard to distinguish failed production from failed survival.

Reviewer #4 (Remarks to the Author):

I believe the authors have adequately addressed all the reviewer comments and the manuscript is now acceptable for publication. For what it's worth, I'm not persuaded by the authors' rebuttal that O-GlcNAcylation of Lyn is likely to be important in the B cell physiology of humans or other mammals, since all data in the manuscript arise from mice, and since S19 of Lyn is poorly conserved across mammals (even other rodents). To be more parsimonious, it might be best to change the sentence on page 19 to read "Nevertheless, our result suggests that O-GlcNAcylation of murine Lyn is critical for the interaction with Syk and activation of Lyn in BCR activation, which may contribute to the homeostasis and activation of B cells in mice and perhaps other organisms."

Reviewer #3:

The main points of the manuscript are, that in the absence of O-GlcNAcylation there is:

- *Increased apoptosis of mature B cells*
- *Decreased signalling of B cells after BCR ligation*
- *OGN-Acylation of Lyn is crucial for its activation and Syk interaction*
- *Increased apoptosis of GC B cells, memory B cells and plasma cells*

The apoptosis conclusion is interesting but I think more complex than is apparent from reading the manuscript and this is why I had asked for the BrdU labeling experiment, which has now been done. The aspects which I find puzzling are that it is restricted to the recirculating B cell compartment in the bone marrow, at least statistically, so it may not be so straightforward to say that apoptosis is the cause of the peripheral B cell deficiency. That is the reason for the BrdU experiment, which has been done, but on the bone marrow B cells. The data presented will require some explanation by the authors to put them into the context of an apoptosis-only model as they suggest a 66% reduced production of mature BM B cells in the KO within the 12 hour pulse. Wouldn't one have expected increased proportional production in the KO compartment if there was increased apoptosis but similar input? At the moment, there is both increased apoptosis and potentially decreased input, meaning that both could be responsible for the deficit in the population. Equally, there are no data turnover for the peripheral B cell populations.

Reply:

We appreciate the thorough evaluation by the reviewer. Our point-by-point responses are provided below.

We analyzed the BrdU labeling data (Figure 2c) as follows: [1] gating of "intact" cells, to eliminate small cell debris, based on forward and side scatters; [2] further gating of B220⁺CD43⁻ cells, including late pre-B⁺, immature B⁺ and mature B cells; [3] further gating of pre-B (IgM^{lo}IgD^{lo}), immature B (IgM^{hi}IgD^{lo-int}), and mature B (IgM^{int-hi}IgD^{hi}) cells. Figure 2c shows that the proportion of BrdU⁺ cells in mature B cells in B-KO mice is lower than that in wild-type mice. On the other hand, the proportion of BrdU⁺ cells in immature B cells in B-KO mice (9.6 ± 1.5 %) is similar to that in wild-type mice (10.3 ± 0.66 %) (table below and panel A of attached figure, which shows representative flow cytometry analysis data from one of three pairs of mice and statistical results from three pairs of mice).

"Intact cell" gate	BrdU ⁺ (%), B-KO	BrdU ⁺ (%), Ctrl	P (Students' t test)
Immature B	9.6 ± 1.5	10.3 ± 0.66	0.69
Mature B	1.5 ± 0.23	4.2 ± 0.49	0.0077

This result can be interpreted in two ways: either (A) B-KO mature B cells failed to proliferate, or (B) B-KO mature B cells proliferated but failed to survive. To distinguish these two alternative possibilities, it is ideal to compare the proportions of BrdU⁺ cells in live cells alone vs. live+dead cells combined. However, it is impossible to combine the BrdU analysis (which requires cell permeabilization) with dye-based live/dead cell discrimination. Therefore, we calculated the proportion of BrdU⁺ cells without the first gating (i.e., exclusion of small cells and debris). The result (table below and panel B of attached figure, which shows representative flow cytometry analysis data from one of three pairs of mice and statistical results from three pairs of mice)

implies that B-KO mature B cells incorporate BrdU (i.e., proliferate) as eagerly as wild-type mature B cells.

Total cell	BrdU ⁺ (%), B-KO	BrdU ⁺ (%), Ctrl	P (Students' t test)
Immature B	13.1 ± 1.4	9.5 ± 0.29	0.059
Mature B	6.6 ± 1.6	4.9 ± 0.68	0.38

A. Intact cells

B. All cells

These results, combined with the data presented in Figure 1 indicating that there is no increase in the number of immature B cells in B-KO mice (Figure 1d), suggest that there should be no developmental block at the stage from immature to mature B cells in B-KO mice. To clarify our point in this aspect, we added another sentence “In theory, if the reduced number of mature B cells is due to the developmental block, the number of immature or transitional B cells should be increased. However, B-KO mice showed normal number of immature and transitional B cells” on page 7. Thus, we conclude that the reduced number of mature B cells in B-KO mice is primarily attributable to the enhanced apoptosis of mature B cells.

Finally, and I apologize for raising this now but I don't think any of the data presented in either the first or revised submission address this point, I think the abstract has to be modified to remove the claim of increased apoptosis in plasma cells in the KO. No evidence was presented on this, I don't think, and at the moment, it is hard to distinguish failed production from failed survival.

Reply:

We apologize for this oversight. We removed phrase "and plasma cells" from the abstract. The revised sentence in abstract is "Ogt deficiency in germinal centre (GC) B cells also results in enhanced apoptosis of GC B cells and memory B cells in an immune response, consequently causing a reduction of antibody levels". We thank reviewer for noting this imprecise statement.

Reviewer #4:

I believe the authors have adequately addressed all the reviewer comments and the manuscript is now acceptable for publication. For what it's worth, I'm not persuaded by the authors' rebuttal that O-GlcNAcylation of Lyn is likely to be important in the B cell physiology of humans or other mammals, since all data in the manuscript arise from mice, and since S19 of Lyn is poorly conserved across mammals (even other rodents). To be more parsimonious, it might be best to change the sentence on page 19 to read "Nevertheless, our result suggests that O-GlcNAcylation of murine Lyn is critical for the interaction with Syk and activation of Lyn in BCR activation, which may contribute to the homeostasis and activation of B cells in mice and perhaps other organisms."

Reply:

We thank the reviewer for the kind suggestion. We amended the sentence as suggested by the reviewer.